# EGFR controls transcriptional and metabolic rewiring in KRAS^G12D colorectal cancer

Dana Krauß [1], Veronica Moreno-Viedma [1], Emi Adachi-Fernandez[1], Cristiano de Sá Fernandes [1,7], Jakob-Wendelin Genger[2,3], Ourania Fari[1], Bernadette Blauensteiner[1], Dominik Kirchhofer [1], Nikolina Bradaric [4], Valeriya Gushchina [1], Georgios Fotakis [5], Thomas Mohr [1], Ifat Abramovich[6], Inbal Mor [6,8], Martin Holcmann [1], Andreas Bergthaler[2,3], Arvand Haschemi[4], Zlatko Trajanoski [5], Juliane Winkler[1], Eyal Gottlieb[6,9] & Maria Sibilia [1]✉

## Abstract

**Inhibition of the epidermal growth factor receptor (EGFR) shows clinical benefit in metastatic colorectal cancer (CRC) patients, but KRAS-mutations are known to confer resistance. However, recent reports highlight EGFR as a crucial target to be co-inhibited with RAS inhibitors for effective treatment of KRAS mutant CRC. Here, we investigated the tumor cell-intrinsic contribution of EGFR in KRAS^G12D tumors by establishing murine CRC organoids with key CRC mutations (KRAS, APC, TP53) and inducible EGFR deletion. Metabolomic, transcriptomic, and scRNA-analyses revealed that EGFR deletion in KRAS-mutant organoids reduced their phenotypic heterogeneity and activated a distinct cancer-stem-cell/WNT signature associated with reduced cell size and downregulation of major signaling cascades like MAPK, PI3K, and ErbB. This was accompanied by metabolic rewiring with a decrease in glycolytic routing and increased anaplerotic glutaminolysis. Mechanistically, following EGFR loss, Smoc2 was identified as a key upregulated target mediating these phenotypes that could be rescued upon additional Smoc2 deletion. Validation in patient-datasets revealed that the identified signature is associated with better overall survival of RAS mutant CRC patients possibly allowing to predict therapy responses in patients.**

**Keywords** EGFR; KRAS; CRC-organoids; Metabolism; Stemness-WNT
**Subject Categories** Cancer; Chromatin, Transcription & Genomics; Digestive System

## Introduction

Colorectal cancer (CRC) ranks as the third most frequent cancer diagnosis and the second leading cause of cancer-related deaths, accounting for 10% of newly diagnosed cancer cases worldwide each year (Siegel et al, 2019; Morgan et al, 2023). It is the second most prevalent cancer in women and the third most prevalent in men (Bray et al, 2018), and there has been a recent rise in incidence in younger adults across genders (Siegel et al, 2022).

CRC development involves the loss of tumor suppressors and accumulation of driver mutations in key signaling pathways like APC, KRAS, TP53, SMAD4 and TGFBR2 (Fearon, 2011; Vogelstein et al, 2013; Li et al, 2021a). In most cases, only two or three driver mutations are necessary for disease progression (Álvarez-Varela et al, 2022; Cancer Genome Atlas Research Network, 2012). One of the most common mutations, *KRAS*, which is present in 43% of CRC cases is associated with a worse prognosis (Hofmann et al, 2022; Nusrat and Yaeger, 2023). Throughout the last decade, KRAS biology underwent enormous investigation, gradually transforming the once considered "un-druggable" oncogene into a target of precision medicine (Zhu et al, 2022).

The epidermal growth factor receptor (EGFR) is a receptor tyrosine kinase involved in many human cancers. In metastatic CRC it is effectively targeted in combination with chemotherapy, leading to significant clinical benefits in patients wild-type for KRAS and BRAF (Fornasier et al, 2018; Bai et al, 2019). Only 50% of patients eligible for anti-EGFR therapy respond to treatment, while the rest display primary resistance for reasons that are yet unknown (Amado et al, 2008; Karapetis et al, 2008; Cox et al, 2014; Brandt et al, 2019). EGFR inhibition was considered ineffective in KRAS mutated patients (Van Cutsem et al, 2009). However, recent studies have shown that EGFR acts as an amplifier of mitogen-activated-protein-kinases (MAPK) signaling in KRAS and BRAF mutant tumors, establishing EGFR as an indispensable treatment component for KRAS^mt CRC (McFall et al, 2019; Ponsioen et al, 2021).

[1]Center for Cancer Research, Medical University of Vienna and Comprehensive Cancer Center, Vienna, Austria. [2]Institute of Hygiene and Applied Immunology, Department of Pathophysiology, Infectiology and Immunology, Medical University of Vienna, Vienna 1090, Austria. [3]CeMM Research Center for Molecular Medicine of the Austrian Academy of Sciences, Vienna 1090, Austria. [4]Department of Laboratory Medicine, Medical University of Vienna, 1090 Vienna, Austria. [5]Biocenter, Institute of Bioinformatics, Medical University of Innsbruck, Innsbruck, Austria. [6]Department of Cell Biology and Cancer Science, The Ruth and Bruce Rappaport Faculty of Medicine, Technion–Israel Institute of Technology, Haifa, Israel. [7]Present address: CeMM Research Center for Molecular Medicine of the Austrian Academy of Sciences, Vienna 1090, Austria. [8]Present address: Department of Molecular Biology, Ariel University, Ariel 4070000, Israel. [9]Present address: Department of Cancer Biology, University of Texas MD Anderson Cancer Center, Houston, TX, USA. ✉E-mail: Maria.Sibilia@meduniwien.ac.at

Also, previous reports in genetically engineered mouse models (GEMMs) of mutant KRAS-driven lung (Moll et al, 2018) and pancreatic tumors (Ardito et al, 2012; Navas et al, 2012) or squamous cell carcinomas driven by the RAS activator Son of Sevenless (SOS) (Sibilia et al, 2000) had already demonstrated that loss of EGFR reduces tumor growth, supporting the hypothesis that EGFR signaling might also act independently of mutant KRAS in tumors. The necessity to include upstream EGFR inhibition to target KRAS mutant pancreatic organoids was recently reported and associated with allele-specific genetic *Kras* mutants, showing that cells only respond to targeting of KRAS$^{G12C}$ when EGFR is concomitantly inhibited (Zafra et al, 2020). Moreover, KRAS inhibitor studies demonstrate EGFR to be essential for synthetic lethal action in combination with the novel non-covalent KRAS$^{G12D}$ MRTX1133 inhibitor (Hallin et al, 2022; Feng et al, 2023; Kataoka et al, 2023).

Parallel to its tumor-intrinsic effects and in the context of CRC and hepatocellular carcinoma (HCC), EGFR was demonstrated to be tumorigenic when expressed in tumor-associated myeloid cells (Lanaya et al, 2014; Hardbower et al, 2017; Srivatsa et al, 2017). GEMMs with EGFR deletion in myeloid cells showed a reduction in CRC and HCC development, while EGFR deletion in KRAS$^{wt}$ tumor cells did not reduce tumor growth. Based on these findings, the precise extent of EGFR's epithelial contribution to CRC development is still debatable and its effect on the tumor microenvironment and the specific underlying resistance mechanisms are still not fully understood.

Since EGFR is a central regulator of cellular physiology, it is not surprising that it is relevant for cellular metabolism. In a cancer setting, EGFR controls several metabolic processes ranging from fatty acid to pyrimidine synthesis, but it is mainly described in the context of the Warburg effect (Orofiamma et al, 2022). In lung adenocarcinoma, EGFR was shown to fuel aerobic glycolysis by maintaining GLUT1 localized to the membrane via PI3K-AKT-mTOR signaling (Makinoshima et al, 2015). EGFR signaling in breast cancer cells was shown to increase aerobic glycolysis by upregulating expression of glycolytic enzymes such as hexokinase 1 (HK1) or phosphorylation of pyruvate kinase M2 (PKM2) (Jung et al, 2019). In prostate cancer, EGFR was also shown to interact and stabilize the active glucose transporter, sodium/glucose co-transporter 1 (SGLT1), independent of its kinase-dependent function (Ren et al, 2013). The metabolic changes induced by EGFR inhibition in CRC and their effect on the tumor microenvironment have so far been poorly studied.

By multi-omics analyses, in this work, we demonstrate that EGFR deletion in KRAS$^{G12D}$ mutant CRC organoids carrying the driver mutations p53 and APC induces metabolic shifts in nutrient selection, alters cell size, differentiation trajectories and is phenotypically evidenced by a distinct WNT and cancer stem cell program driven by the key mediator SMOC2.

# Results

## EGFR deletion in KRAS$^{G12D}$ tumor organoids does not affect proliferation but alters metabolite uptake and secretion

To assess the cell-intrinsic role of EGFR in KRAS$^{G12D}$ mutant tumor cells without the surrounding stroma and immune cells, we generated

murine CRC organoids isolated from colonic tumors of GEMMs harboring an APC mutation ($APC^{min/+}$) and a p53 deletion in intestinal epithelial cells (AP organoids) (Figs. 1A and EV1A). In addition, some organoids carried the $KRAS^{G12D}$ oncogene (AKP organoids) and floxed EGFR alleles (Figs. 1A and EV1A). Recombination of the floxed EGFR allele was induced through Tamoxifen-inducible Cre activation or via Adeno-Cre infection in AKP organoids (Tamoxifen (4-OHT)/Adeno-Cre), resulting in paired AKP and AKPE (EGFR knockout) organoids (Fig. 1A). Cre recombination activity was assessed with a dual-reporter cassette, which also facilitated clonal enrichment by FACS-sorting (Fig. EV1B) (Schönhuber et al, 2014). Genotyping (genomic PCR), RT-qPCR and western blot analysis verified the presence or absence of EGFR (Fig EV1C–E). As for the nomenclature, gene deletion or oncogenic activation are designated as A ($Apc^{min/+}$), K ($Kras^{G12D/+}$), P ($Trp53^{\Delta}$), or E ($Egfr^{\Delta}$), resulting in AKP, AKPE and AP combinations (Figs. 1A and EV1A).

As expected and previously described (Drost et al, 2015; Matano et al, 2015), AP (KRAS$^{wt}$) organoids showed decreased proliferation compared to AKP organoids (KRAS$^{G12D}$) (Fig. 1B). However, AKPE (Egfr$^{\Delta}$) organoids did not show differences in doubling times compared to AKP organoids, demonstrating that loss of EGFR does not affect proliferation (Fig. 1B). This was additionally confirmed by flow-cytometry-based quantification of KI67 and EdU-incorporation (Fig. 1C left, right). Additionally, apoptosis assessed by Annexin V staining was similar between organoids of all genotypes (Fig. 1C middle).

Although the effect of KRAS mutations on cellular metabolism has been extensively investigated (Mukhopadhyay et al, 2021), the contribution of EGFR in KRAS mutant organoids has not been described. Therefore, we further compared the two organoid pairs, AKP and AKPE, and assessed whether EGFR deletion would result in altered extracellular uptake and secretion of metabolites. Absolute quantification of extracellular metabolites in the supernatants showed decreased glucose uptake in AKPE compared to AKP organoids (Fig. 1D). In line with this observation, the levels of GLUT1 protein, the major cellular glucose transporter, showed a significant decrease in EGFR-deleted AKP organoids (Fig. 1E), similar to the levels observed in AP organoids which are KRAS wild-type (Fig. EV1F). We also detected slightly reduced levels of Glut1 protein on the cell surface (Fig. 1F). Interestingly, RNA levels of *Glut1/Slc2a1* were similar between AKP and AKPE organoids (Fig. EV1G), indicating that mutant KRAS signaling leads to EGFR-dependent increase of GLUT1 protein levels by a posttranscriptional mechanism.

Furthermore, significantly lower lactate secretion and a simultaneous decrease in pyruvate uptake resulted in a substantial increase in the pyruvate-to-lactate ratio in the supernatant of AKPE organoids (Fig. 1G–I). This ratio represents the relative contribution of mitochondrial versus glycolytic energy sources (Rodríguez-Colman et al, 2017) and thus highlights a decrease in glycolytic activity in AKPE organoids. Glutamate is an output of the TCA cycle as it is generated from the intermediate alpha-ketoglutarate (aKG). Glutamate secretion was significantly decreased, while glutamine uptake was not altered in AKPE organoids (Fig. 1J,K). Collectively, EGFR deletion led to an altered uptake and secretion of glucose, lactate, pyruvate, and glutamate.

## EGFR deletion fuels glutamine anaplerotic diversion

Metabolic cues are described to maintain oncogenic phenotypes and are recognized as a major hallmark of cancer (Pavlova et al, 2022).

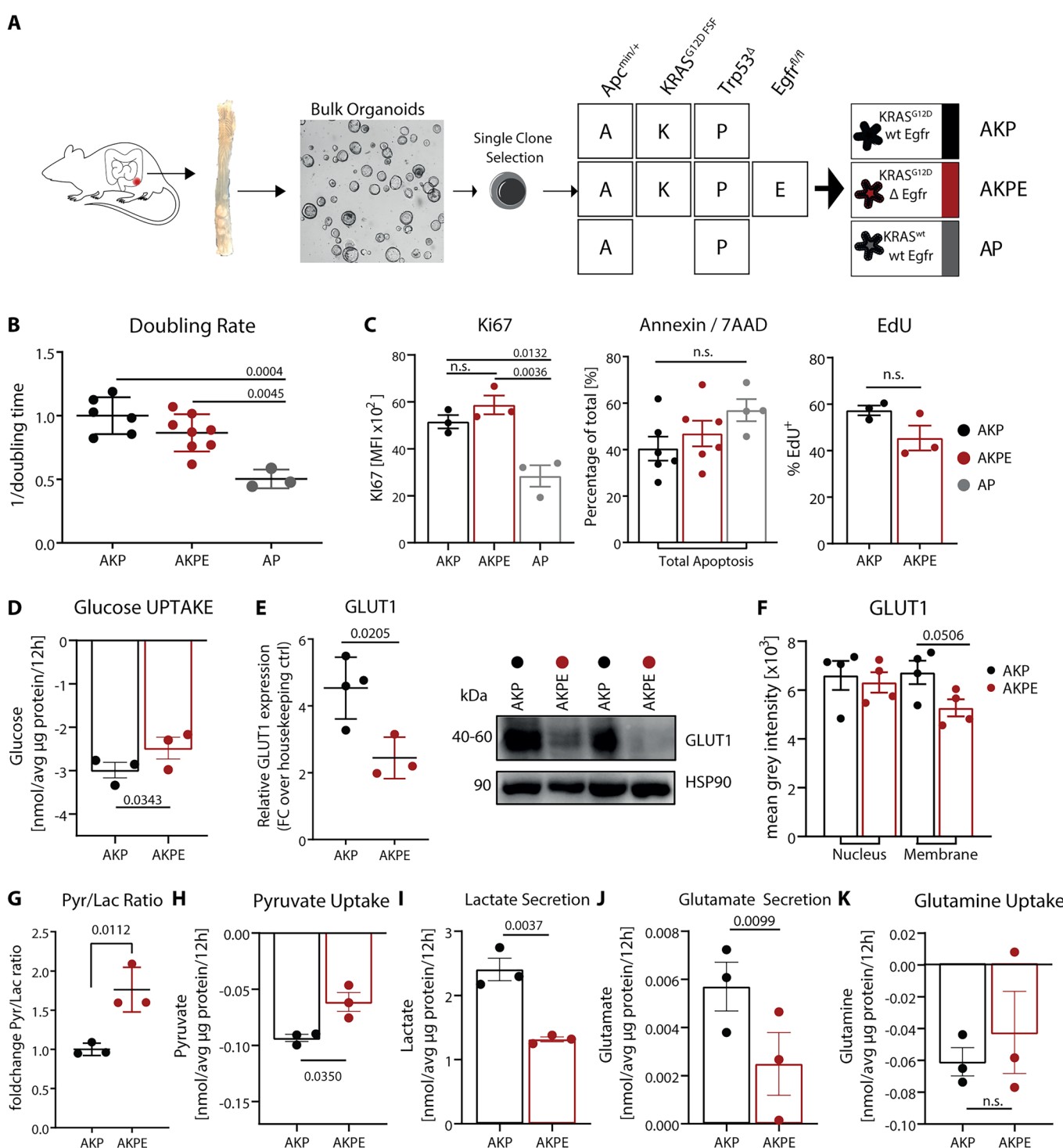

Intrigued by our observation of differential extracellular metabolite profile, we further set out to delineate underlying metabolic changes of EGFR signaling in KRAS[G12D] mutant organoids. We hypothesized that anaplerotic glutaminolysis could compensate for decreased glucose uptake and utilization in AKPE organoids. Therefore, we selected stable isotope tracer U-[13]C-glutamine followed by liquid chromatography–mass spectrometry (LC-MS)-based metabolomics analysis of AKP and AKPE organoids at basal conditions. Tracing

isotopically labeled metabolites allows us to assess the contribution of a distinct carbon source to different pathways. In addition, it provides information about the differential contribution to the percentage labeling of respective metabolites. We cultured organoids with stable U-[13]C-glutamine isotope for eight hours and measured the incorporation of [13]C into intracellular or extracellular metabolites using an in-house curated library, which covers central carbon metabolic pathways including glycolysis and the TCA cycle (Fig. 2A).

**Figure 1. EGFR deletion in KRAS$^{G12D}$ tumor organoids does not affect proliferation but alters metabolite uptake and secretion.**

(A) Schematic representation of intestinal-specific CRC organoid generation from GEMMs harboring deletion or oncogene activation in A ($Apc^{min/+}$), K ($Kras^{G12D/+}$), P ($Trp53^\Delta$), or floxed E ($Egfr^{fl/fl}$) alleles, resulting in AKP, AKPE and AP organoids. (B) Doubling rate of AKP, AKPE, and AP organoids ($n = 6$, 8 or 3, respectively). (C) Flow cytometry analysis of mean fluorescence intensity of Ki67 ($n = 3$) or percentage of early (AnnexinV +) and late (Annexin V + and 7AAD +) apoptotic ($n = 6$, 6 or 4) or EdU-positive ($n = 3$) cells of AKP, AKPE, and AP organoids. (D) Quantification of absolute glucose levels in supernatant by liquid chromatography–mass spectrometry (LC-MS) analysis ($n = 3$). (E) Quantification of absolute GLUT1 protein amount between AKP and AKPE organoids ($n = 4$, 3) (left). GLUT1 protein levels assessed by western blot (right). (F) Mean gray intensity GLUT1 expression levels in membrane or nuclear compartments assessed from immunofluorescence-stained organoids. (G) Paired pyruvate over lactate ratio of respective absolute quantities ($n = 3$). (H–K) Quantification of absolute pyruvate, lactate, glutamate and glutamine levels in supernatant by LC-MS analysis ($n = 3$). All data represent mean $+/-$ SEM of at least three biologically independent organoids. $P$-values calculated by paired, two-tailed t-test (between pairs of AKP and AKPE organoids) or one-way ANOVA. Source data are available online for this figure.

Analysis of the labeled percentage for each metabolite demonstrated that the overall incorporation of labeled carbons from uniformly labeled glutamine was substantially increased in AKPE organoids (Figs. 2B and EV2A,B, Dataset EV1). Specific analysis of TCA cycle metabolites demonstrated an increase in the total levels of aKG, succinic acid, fumarate, and malate in AKPE organoids (Fig. 2C). Moreover, all analyzed TCA intermediates also had elevated incorporation of labeled carbons from glutamine, namely the $M + 4$ and $M + 5$ isotopologues in AKPE organoids (Figs. 2C and EV2B). Cis-aconitate and citrate demonstrated an appreciable fraction of the $M + 4$ isotopologues (Fig. EV2B), further showing that glutamine contributes to labeling of TCA intermediates through oxidative cycling. These results indicate that organoids with EGFR deletion had an increased usage of glutamine for anaplerotic metabolism.

We have shown decreased GLUT1 (Fig. 1E) expression, as well as decreased glucose uptake in AKPE organoids (Fig. 1D). We therefore performed a glucose tracing experiment with U-$^{13}$C glucose. This was done at early (15 min) and late (eight hours) timepoints, with the aim of assessing the immediate and downstream incorporation of labeled glucose (Fig. EV2C). Inside the cell, glucose is rapidly phosphorylated by hexokinases. We observed that total levels of proximal glycolytic intermediates (G6P: glucose-6-phosphate and F6P: fructose-6-phosphate) were decreased in AKPE organoids (Fig. EV2D), corroborating decreased glucose uptake in AKPE organoids (Fig. 1D). Overall, our metabolomics tracing experiments revealed that EGFR is needed to maintain glucose incorporation into glycolytic metabolites and thus EGFR deletion resulted in a decrease of glucose usage and simultaneous increase toward glutamine utilization into the TCA.

To functionally assess the metabolic phenotype, we used an extracellular flux analyzer to measure oxygen consumption rate (OCR) as a surrogate marker for mitochondrial respiration. AKPE organoids exhibited higher OCR at basal state and maximal respiration when compared to AKP organoids (Fig. 2D). Extracellular acidification rates (ECAR) measuring glycolytic activity revealed no significant difference between AKP and AKPE organoids (Fig. EV2E). In addition, we selected a set of metabolic enzymes involved in major metabolic nodes to assess enzymatic activities that would contribute to the different phenotype between AKPE and AKP organoids. Determining the mean enzyme activity per cell, we observed a decrease in glyceraldehyde 3-phosphate dehydrogenase (GAPDH; glycolysis) and glucose-6-phosphate dehydrogenase (G6PDH; PPP) activity (Fig. 2E), reflecting our isotope-tracing results. In summary, these data corroborated observed shifts in nutrient utilization (Fig. 1), highlighting and reinforcing decreased glucose utilization and

enhanced glutaminolysis upon EGFR deletion in AKPE organoids (Fig. 2F).

## EGFR deletion invokes a distinct transcriptional signature in KRAS$^{G12D}$ cells

To better understand the effect of EGFR deletion in KRAS$^{G12D}$ CRC organoids, we additionally performed bulk RNA sequencing of AKP and AKPE organoids. We also included wild-type KRAS (AP) tumor organoids to be able to distinguish the changes induced by mutant KRAS. Overall, transcriptional profiles of the respective genotypes revealed clear separation in principal component analysis (Fig. 3A). This indicates the reproducibility of our biological organoid replicates and highlights solid differences in gene expression across genotypes.

As expected, a large number of genes were differentially regulated between AKP and AP organoids (4968 genes). Moreover, AKPE organoids showed a specific subset of genes that are solely upregulated upon EGFR deletion but are not affected by KRAS mutational status, showing a distinct effect of EGFR, in the presence of the KRAS$^{G12D}$ mutation (Fig. 3B). In fact, AKPE exhibits 1538 genes differentially expressed in comparison to AKP organoids (Fig. 3B,C). Among those, genes such as *Smoc2*, *Ascl2*, and *H6pd* exhibit the highest upregulation and importantly *Egfr* showed the highest negative fold-change (Fig. 3C), further evidencing successful EGFR deletion. Consistent with the observed increased proliferation in AKP in comparison to AP organoids (Fig. EV3A) genes associated with cell cycle progression were upregulated but not altered between AKP and AKPE organoids (Fig. 1B,C). Transcripts of other ErbB receptor tyrosine kinases such as *Erbb2* and *Erbb3* remained unaffected (Fig. EV3B). Interestingly, EGFR ligands, such as *Areg*, *Btc*, *Ereg*, *Tgfa* and *Hbegf* were downregulated (Fig. EV3B) suggesting that a possible autocrine stimulation loop was impaired in the absence of EGFR in tumor cells. Together, this highlights that the presence of EGFR has a significant impact on the transcriptional profile of cancer cells despite the existing KRAS$^{G12D}$ activating mutation.

As we demonstrated EGFR-induced metabolic alterations in the AKPE organoids compared to AKP, we investigated whether metabolic changes were also reflected at the transcriptional level. We selected curated metabolic genes established by the RECON3 consortium (Brunk et al, 2018) and surprisingly, unbiased hierarchical clustering analysis of the metabolic gene subsets resulted in the clustering of AKPE organoids closer together with AP organoids (Figs. 3D and EV3C,D). Categorizing differentially expressed genes according to their functional role primarily showed alterations in the pentose phosphate pathway, glycolysis, glutamine

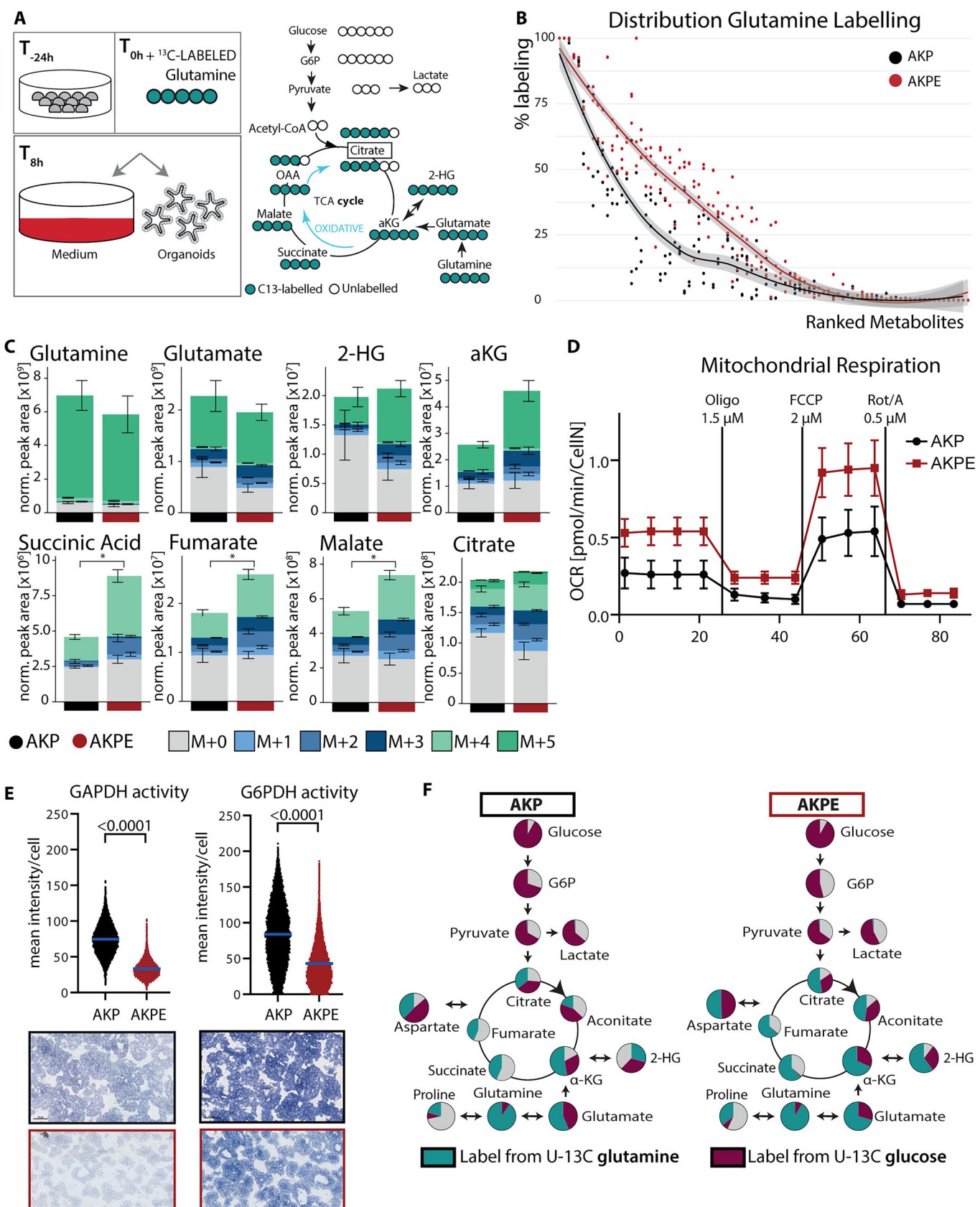

◄ **Figure 2. EGFR deletion fuels glutamine anaplerotic diversion.**

(A) Experimental design and schematic overview of stable isotope tracing metabolomics studies of AKPE and AKP cultures with U-$^{13}$C-glutamine for 8 h and schematic overview of glycolytic, oxidative glutamine catabolism and fractional contribution of labeled carbons of indicated metabolites. (B) Dot plot of fractional labeling from glutamine of respective metabolites. List of ranked metabolites according to percentage $^{13}$C-glutamine labeling provided as Dataset EV1. (C) Glutamine tracer metabolomics in AKPE ($n = 3$, biological) or paired AKP ($n = 3$, biological) organoids (technical replicates 4 per organoid line) showing absolute abundance and fractional labeling of glutamine, glutamate, 2-hydroxyglutarate (2-HG), alpha-ketoglutarate (aKG), succinic acid, fumarate, malate and citrate. M + 0 (all carbons unlabeled) to M + $n$ isotopologues indicate number of $^{13}$C atoms present in respective metabolite. Total abundance normalized to protein (BCA) content. (D) Representative oxygen consumption rate (OCR) measurement of AKP and AKPE organoids ($n = 3$, biological) obtained by the Mito Stress test of the Seahorse XF analysis. Oligomycin, FCCP, Rotenone (Rot) and antimycin-A (A) were added at indicated timepoints. (E) Quantification of mean intensity per cell enzyme activity of glucose-6-phosphate dehydrogenase (G6PD) and glyceraldehyde 3-phosphate dehydrogenase (GAPDH) and representative images of one representative biological replicate. Scale bars, 50 µm. (F) Simplified representation of glycolysis, TCA and glutaminolysis. Area of pie-charts represents percentage of abundance of respective metabolites. Purple, turquoise, gray represent fractional labeling coming from U-13C glucose, U-13C glutamine or other sources, respectively. aKG alpha-ketoglutarate, G6P Glucose-6-phosphate, 2-HG 2-hydroxyglutarate, TCA tricarboxylic acid. All data represent mean $+/-$ SEM. P-values calculated by paired, two-tailed t-test (between pairs of AKP and AKPE organoids). Source data are available online for this figure.

metabolism, and upregulation of TCA/OXPHOS genes after EGFR deletion (Fig. EV3E). Further, gene set enrichment analysis (GSEA) of differentially expressed genes of AKPE versus AKP unbiasedly revealed pathways associated with metabolism such as the oxidative phosphorylation, glycolysis or fatty acid metabolism, hence confirming our results obtained with the metabolomic analysis (Fig. EV3F).

As we aimed to understand underlying mechanisms driving the metabolic phenotype, we noted downregulation of additional pathways such as MAPK, ErbB, JAK-STAT, and RNA biogenesis, which are known pathways activated by autocrine EGFR stimulation in accordance with observed downregulation of EGFR ligands (Fig. EV3B) (Sigismund et al, 2018). Interestingly, there was also upregulation of WNT signaling pathways in AKPE organoids compared to AKP (Fig. 3E). Despite the oncogenic KRAS$^{G12D}$ mutation, EGFR deletion in organoids resulted in a distinct transcriptional signature, confirming our observed metabolic changes and additionally uncovering upregulation of autocrine signaling and WNT pathways.

## EGFR deletion results in distinct WNT and stemness transcriptional program

Given the metabolic and transcriptional alterations invoked by EGFR deletion in a KRAS$^{G12D}$ mutant background, we next investigated the possible underlying mechanism by mining our transcriptomic dataset. For this purpose, we investigated genes that are uniquely regulated through EGFR by intersecting changes resulting from comparisons of AKPE versus AKP or AP versus AKP and identified genes that are exclusively upregulated in AKPE organoids (Figs. 4A and EV4A). Categorizing resulting genes showed association with known cellular and molecular functions of EGFR like cell motility, adhesion, cell fate, differentiation, and signaling. Notably, specific sets of metabolic and WNT pathways were upregulated in AKPE organoids (Fig. 4A).

In order to corroborate the previous observations, we assessed activity inference of signaling pathways through analysis of pathway response genes by PROGENy (Pathway RespOnsive GENes) (Schubert et al, 2018). We compared AKP to AKPE and included AP organoids as a reference to delineate EGFR-specific effects from KRAS mutation-invoked alterations. AKPE organoids, although carrying KRAS$^{G12D}$ mutations, displayed downregulated PI3K, EGFR, and MAPK pathway signatures similar to those in AP

organoids (Fig. 4B). Other pathways, such as NFkB, TNFa, TGFb, or JAK-STAT remained unaffected in AKPE organoids but were upregulated in AP organoids. Surprisingly, only the WNT pathway response genes were upregulated in AKPE organoids compared to AKP and AP (Fig. 4B). Observing this distinct signature, we hypothesized that the presence or absence of EGFR in KRAS$^{G12D}$ organoids mainly influences the WNT pathway. Indeed, EGFR-dependent reprogramming of KRAS mutant organoids upregulated not only genes directly involved in WNT signaling but also response genes were upregulated in AKPE organoids (Fig. 4B).

Physiological WNT signaling is closely associated with somatic and stem cell maintenance (Ring et al, 2014; Gehart and Clevers, 2019), and CRC tumors are known to depend on aberrant WNT signaling to maintain their stemness and de-differentiation phenotype (Kleeman and Leedham, 2020). As APC$^{mut}$ adenomas are independent of WNT ligands (Kleeman et al, 2020), we were intrigued by the distinct and unique effect of EGFR deletion on WNT pathway response genes present in APC-mutant background.

To further address this observation, we manually curated WNT, intestinal stem cell (ISC), and cancer stem cell (CSC) pathway signatures of numerous recent publications (Table EV1) and performed gene set variation analysis (GSVA) on AKPE organoids. We observed a specific pathway enrichment of particular WNT and stemness classified pathways, divergent to those of AP organoids (Fig. EV4B). To identify and separate the unique WNT and stemness signatures associated with EGFR deletion from KRAS effects, we performed weighted gene correlation network analysis (WGCNA) on the constituting genes of these curated pathways. Thus, we identified twelve significant module eigengenes, of which five and seven modules were significantly regulated at the pathway level (Fig. EV4C,D). We stratified data according to these modules and underlying genes exclusively associated with EGFR deletion (Figs. 4C and EV4D: module four, nine) and identified an EGFR-specific gene signature associated with WNT and stem cell signaling (AKPE Signature) (Fig. 4D,E).

Among those genes, *Smoc2* and *Areg* had the highest significance and fold changes, being up- and downregulated, respectively (Fig. 4E). *Smoc2*, the gene with the highest fold-change in our AKPE signature encodes for the extracellular matrix protein SPARC-related modular calcium binding 2 and is associated with stem cell signatures and cell type-specific differentiation (Jang et al, 2020), while *Areg* encodes for amphiregulin, a well-described EGFR ligand (Abud et al, 2021)

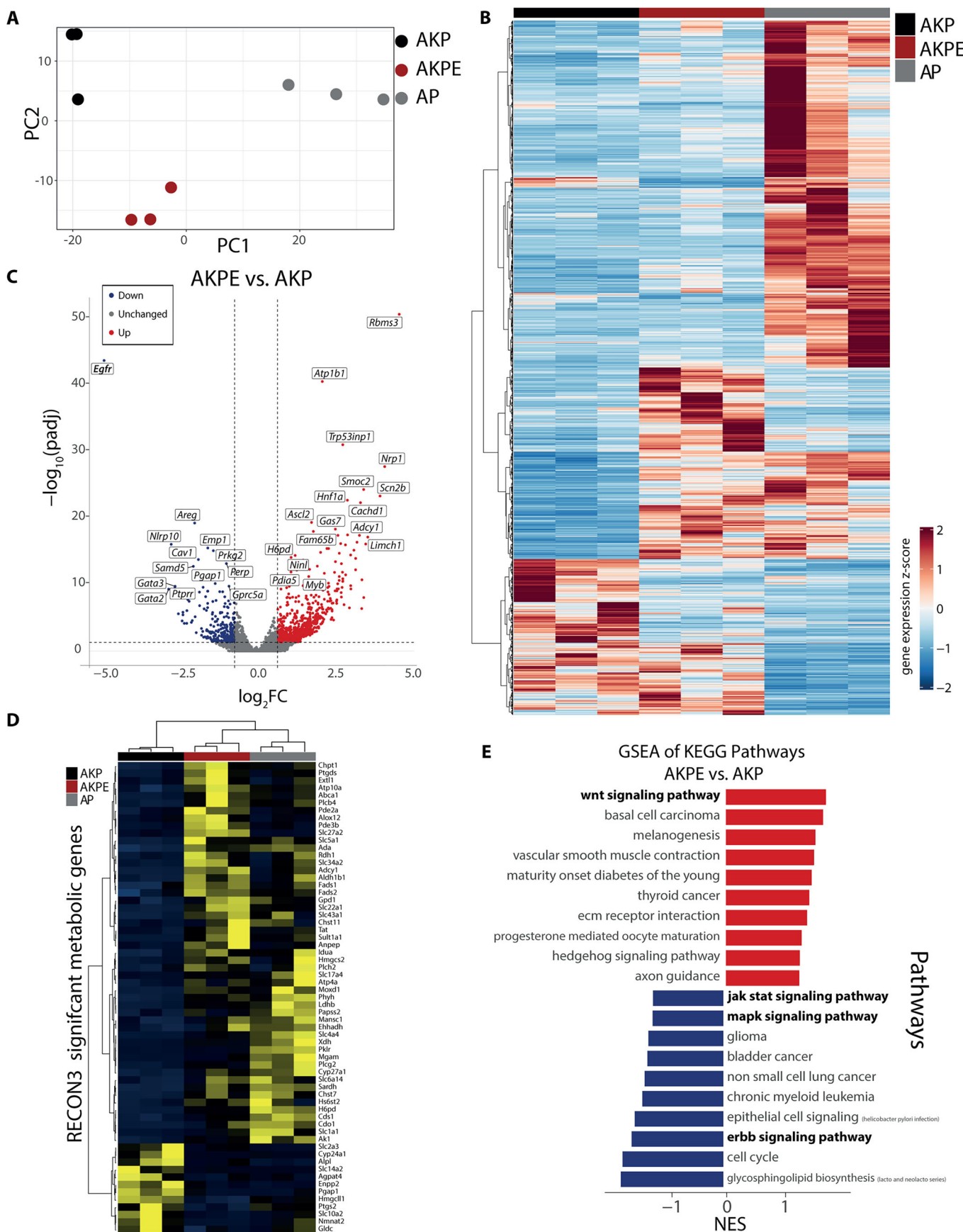

**Figure 3.    EGFR deletion invokes a distinct transcriptional signature in KRAS^G12D cells.**

(**A**) Principal component analysis (PCA) of RNA-sequencing data of AKP, AKPE and AP organoids in steady state. Each subpopulation is depicted by indicated color. (**B**) Heatmap showing expression of differentially expressed genes (DEGs) in AKP, AKPE and AP organoids at steady state. Each column represents one organoid line derived from one mouse. (**C**) Volcano plot representing up or down-regulated gene expression for AKPE versus AKP organoids ($n = 3$). (**D**) Heatmap showing subset of differentially expressed genes of metabolic RECON3 curated gene list in AKP, AKPE and AP organoids. (**E**) Gene set enrichment analysis (GSEA) showing MSigDB KEGG pathways significantly enriched in AKPE versus AKP ranked by their normalized enrichment score (NES).

and represents the most downregulated gene. Other genes known to be associated with stemness such as *Reg4*, *Prox1*, *Vim, Mex3a*, and *Lef1* were found also upregulated (Fig. 4E) (Álvarez-Varela et al, 2022; Heino et al, 2021; Kleeman et al, 2020). GSVA analysis revealed specific upregulation of canonical WNT signaling genes in AKPE organoids (Fig. EV4F). Further probing the CellChat database (Jin et al, 2021) for WNT-ligand-receptor interactions revealed that AKPE organoids showed specific upregulation of canonical WNT signaling regulating receptors (*Fzd2, Fzd9, Fzd10*) and ligands (*Wnt5b, Wnt6, Wnt10b, Wnt16*) (Fig. EV4G) (Zhu and Li, 2023). Taken together, these results demonstrate that the absence of EGFR in KRAS^G12D organoids results in enrichment of a unique WNT- and stemness signature.

We further verified the observed transcriptional WNT response and found that components of the WNT pathway such as β-catenin and LEF1 were also upregulated at the protein level (Fig. 4F,G). Previously, *Smoc2* was shown to drive a CSC-specific signature predicting better clinical outcomes in CRC patients (Jang et al, 2020). According to the increased transcriptional expression, SMOC2 protein levels were also increased in KRAS mutant organoids lacking EGFR (Fig. 4H). To investigate if EGFR signaling negatively regulates *Smoc2* expression, we performed a time course analysis and monitored if, upon EGFR deletion in KRAS mutant organoids, *Smoc2* expression was upregulated. Expression of *Smoc2* was followed upon tamoxifen (4-OHT) induced deletion of *Egfr*. Tamoxifen administration reduced *Egfr* expression by 50% to 83% after 24 h and 48 h, while *Smoc2* expression increased by 1.7x to 7x-times at the respective timepoints (Fig. 4I). Importantly, a similar increase was observed by Erlotinib treatment after 24 h (Fig. EV4H). At the same time, *Lef1*, a WNT-downstream effector, only increased marginally throughout the time course, indicating that *Smoc2* expression precedes induction of WNT target genes at a transcriptional level (Figs. 4I and EV4H). We observed an increase of *Smoc2* expression after EGFR deletion in an independent metastatic AKPS CRC organoid model where SMAD4 is additionally deleted (AKPS/AKPSE), highlighting the role of EGFR as a negative regulator of SMOC2 and WNT signaling pathway (Fig. EV4I,J).

## EGFR deletion alters cell size, while Smoc2 knockout rescues all phenotypes induced by EGFR loss

WNT/β-Catenin signaling was recently identified as the driver of volumetric compression of individual ISCs that mediates intracellular crowding in intestinal organoids (Li et al, 2021b). We therefore analyzed cell size in AKPE organoids across time. We observed decreased circumference and diameter in AKPE when compared to AKP organoids (Fig. 5A), which is surprising given the presence of the oncogenic KRAS mutation. Similar results were obtained by forward scatter analysis (Fig. 5B). Targeting EGFR in

AKP organoids with the small molecule inhibitor Erlotinib also led to smaller cell size (Fig. 5C). To confirm that cell size alteration is linked to EGFR, we also demonstrated reduced cell size in AKPSE cells (Fig. EV5A).

Since stem cells are known to be small in size and expressing also high levels of SMOC2, we hypothesized that the high SMOC2 levels observed in EGFR-deficient organoids could be responsible for cell size reduction. Therefore, we employed the CRISPR/Cas9 system to generate *Smoc2* knockout organoids in both *Egfr* wild-type (AKP) and *Egfr* knockout (AKPE) backgrounds (Fig. EV5B). As shown in Fig. 5D, deletion of *Smoc2* in AKPE organoids completely rescued the cell size reduction, while *Smoc2* deletion in *Egfr* wild-type organoids had no effect on cell size (Fig. 5D).

We next investigated whether SMOC2 was also causally responsible for the other phenotypes induced by *Egfr* deletion and therefore conducted RNA sequencing analysis. As assessed by Euclidean distance, we observed most differences between AKP and AKPE organoids lacking SMOC2 (Fig. 5E). Also, the induction of most WNT and stemness-associated genes of the AKPE-gene signature was reversed when *Smoc2* was deleted in EGFR-deficient organoids and the transcriptional landscape was similar to *Egfr* wild-type (AKP) organoids (Fig. 5F).

The previously observed decrease in glucose uptake, characteristic of EGFR-deficient (AKPE) (Fig. 1D) organoids, was rescued when *Smoc2* was additionally deleted (Fig. 5G). AKPE organoids lacking SMOC2 exhibited glucose uptake levels comparable to *Egfr* wild-type (AKP) organoids, while *Smoc2* deletion did not affect AKP organoids. Analysis of metabolic gene expression revealed that *Smoc2* knockout altered a small subset of genes, e.g., *H6pd*, associated to glucose and PPP-balance or TCA cycle genes as *Pdk2, Sdhd, Idh1, Idh3a* or *Sdhb*, particularly restricted to EGFR-deleted organoids (Fig. 5H). These findings demonstrate that SMOC2 acts as a critical mediator of the transcriptional and metabolic phenotypes driven by the absence of EGFR leading to activation of WNT and stemness signaling.

We next explored whether the metabolic changes observed in EGFR-deficient organoids could be driving the increased WNT and stemness signatures in a reciprocal manner. Therefore, metabolic perturbation experiments were performed by treating AKP and AKPE organoids with BPTES, a glutaminolysis inhibitor that blocks the conversion of glutamine to glutamate. RNA sequencing analysis revealed genotype-specific effects of BPTES treatment, with distinct subsets of TCA cycle, oxidative phosphorylation (OXPHOS), glycolysis, and glutamine metabolism-associated genes being uniquely upregulated in either AKP or AKPE organoids (Fig. EV5C). However, BPTES treatment did not impact the WNT/stemness gene signature specific to EGFR-deleted organoids (Fig. EV5D). These results suggest that the metabolic changes induced by EGFR deletion are not responsible for the induction of WNT and stemness gene signatures.

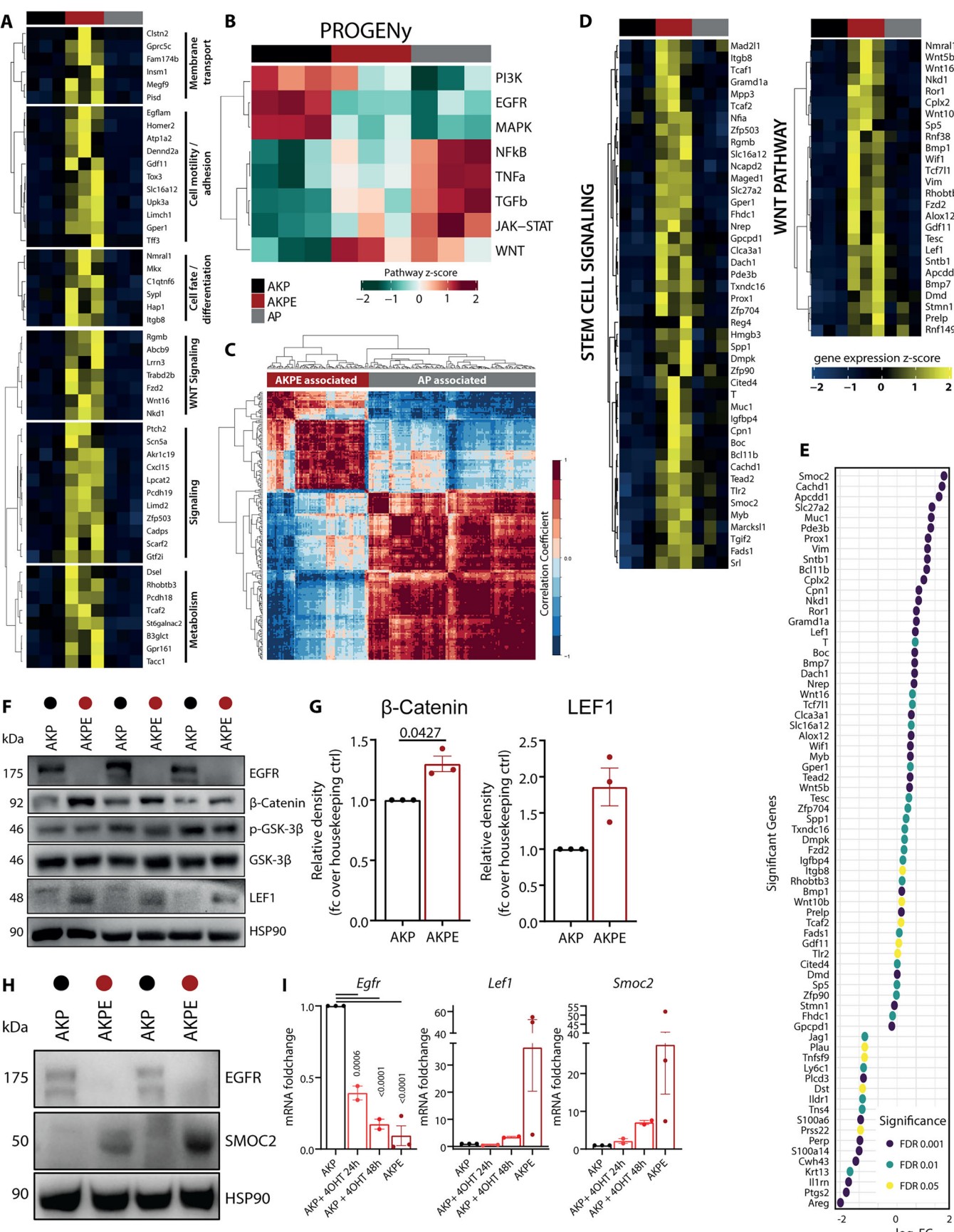

**Figure 4.   EGFR deletion results in distinct WNT and stemness transcriptional program.**

(A) Heatmap showing expression of genes uniquely upregulated in AKPE compared to AKP and AP organoids categorized according to molecular function. (B) Inference analysis of RNA-seq data with PROGENy. Pathway activity scores are z-scaled und hierarchical clustering was performed by Euclidean distance and complete linkage. (C) Hierarchical clustering of the WGCNA derived significant gene modules: AP-associated genes (gray) and AKPE-associated genes (red). (D) Heatmap showing expression of upregulated significant AKPE-associated genes categorized by WNT pathways or stem cell signaling. (E) Dot plot representing up or down-regulated gene expression of AKPE-associated genes according to fold-change, colored by FDR. (F) Westernblot analysis of WNT pathway proteins of the three biologically derived organoids used in this study. (G) Quantification of LEF1 and β-catenin protein expression ($n = 3$). (H) Representative SMOC2 protein expression assessed by western blot. (I) RT-qPCR analysis of *Egfr, Smoc2, Lef1* mRNA expression in AKP, AKPE and 24/48h- 4-OHT treated AKP organoids ($n = 3, 3, 2$ or $2$, respectively). All data represent mean $+/-$ SEM. *P*-values calculated by paired, two-tailed t-test (between pairs of AKP and AKPE organoids) or One-way ANOVA. Source data are available online for this figure.

We therefore investigated whether WNT activation can cause the metabolic phenotypes induced by EGFR deletion in AKPE organoids by inhibiting WNT with the β-catenin/TCF suppressor ICG-001 (ICG) in AKPE organoids. We also tested if activation of WNT signaling in EGFR wild-type (AKP) organoids would phenocopy the metabolic changes observed after EGFR deletion by employing the GSK3β inhibitor CHIR99021 (CHIR). RNA sequencing analysis showed that activation of the WNT signaling pathway with CHIR changed the expression of a subset of metabolic genes in EGFR wild-type (AKP) organoids (e.g., mostly TCA cycle genes as *Sdhb, Idh1, Idh3a*), indicating that WNT signaling can regulate certain metabolic pathways in the presence of EGFR. However, the metabolic phenotype observed in EGFR-deleted organoids could not be replicated simply by activating WNT in EGFR expressing organoids (only four TCA genes showed differential expression; as indicated *Pdk2, Acss1* or *Tcirg1* and *Uqcrc2*) (Fig. EV5E), thus did not prevent the distinct metabolic gene expression patterns observed in EGFR deficient organoids. Similarly, alteration of the WNT/stemness signature, was not altered upon CHIR or ICG treatment in AKP or AKPE organoids (Fig. EV5F). These results demonstrate that the metabolic phenotype induced by EGFR deletion is not exclusively driven by WNT signaling but involves broader EGFR-dependent regulatory networks.

Taken together, SMOC2 emerges as a key mediator of most of the phenotypic alterations observed in CRC organoids lacking EGFR, driving WNT and stemness pathway activation, reduction in cell size as well as metabolic changes.

## EGFR deletion reshapes cellular heterogeneity and differentiation trajectories in KRAS^G12D tumor organoids

Given the established role of WNT signaling in stemness and cellular differentiation, along with our discovery of a distinct WNT/cancer stemness transcriptional program following EGFR deletion, we hypothesized that EGFR deletion could alter the balance between different cell populations, affect the cellular composition and differentiation trajectory of tumor organoids.

To determine whether the EGFR-dependent changes in cell size and transcriptional profiles were linked to shifts in cellular heterogeneity we performed single-cell RNA-sequencing (scRNA-seq) on two AKP and AKPE organoid lines. Unbiased clustering of AKP and AKPE organoids revealed that cells primarily clustered according to genotype, underscoring the distinct transcriptional landscapes driven by EGFR deletion (Fig. 6A). Additionally, AKP organoids show higher cell heterogeneity (Fig. 6B–D), suggesting that EGFR deletion reduces cell population diversity. As scRNA-seq does not reliably capture *Egfr* expression (Fig. 6E) (Fari et al, manuscript in preparation), we confirmed that previously identified

genes of interests from our bulk sequencing as *Ctnnb1* (β-Catenin) and our AKPE signature are recapitulated in the EGFR-deleted organoid cells (Fig. 6E). Notably, while both AKP and AKPE organoids lacked canonical differentiation markers such as *Krt20, Muc2,* and *Dclk1*, we observed distinct differences in the expression and distribution of stemness and proliferation markers across subclusters (Fig. 6F). In AKP organoids, stemness markers were more distinctively expressed across subclusters, with genes like *Ascl2* and *Agr2* highly expressed in subclusters AKP1-3 but reduced in AKP4. Conversely, *Lrig1* and *Lgr5*, known markers of intestinal stem cells, were predominantly enriched in subcluster AKP4, highlighting differences in stemness signatures within AKP organoid cell populations. In contrast, AKPE organoids demonstrated uniformly high expression of *Smoc2* and *Prox1*, two genes absent in AKP organoids, suggesting a transcriptional reprogramming driven by EGFR deletion (Fig. 6E–G). Proliferation marker, *Mki67* was restricted to subclusters AKP3, AKP4, and AKPE3, indicating that not all subclusters within these organoids contribute equally to proliferative activity (Fig. 6F).

To further investigate the differentiation status of the identified cell populations, we performed Cytotrace2 (Kang et al, 2024). Potency score and potency category prediction indicated that both AKP and AKPE organoids show almost no fully differentiated cell types (Fig. 6H,I). However, AKPE organoids exhibit a larger fraction of cells with higher potency and overall show higher potency scores (Fig. 6H,I).

Collectively, these data suggest that there is a shift in cell populations and differentiation status in AKP and AKPE organoids. AKP organoids consist of heterogeneous cell populations that express markers of early stemness and present themselves in less differentiated cell states, whereas AKPE organoids are more homogenous, express later stemness genes and are more differentiated but do not reach fully differentiated cell states. Therefore, EGFR shows a distinct regulatory in cellular composition and differentiation in KRAS mutant organoids.

## EGFR deletion implicates synergistic sensitivity to KRAS^G12D inhibition by MRXT1133

To assess the clinical relevance of our findings, we evaluated whether the increased WNT/stemness (AKPE) signature observed after EGFR loss, would be associated with an increased response to KRAS inhibitors, specifically the selective KRAS^G12D inhibitor MRTX1133 (Wang et al, 2022; Hallin et al, 2022) that is currently under clinical evaluation (NCT05737706). Proliferation assays revealed that MRTX1133 treatment had a minimal impact on the proliferation of AKP organoids, while it completely inhibited

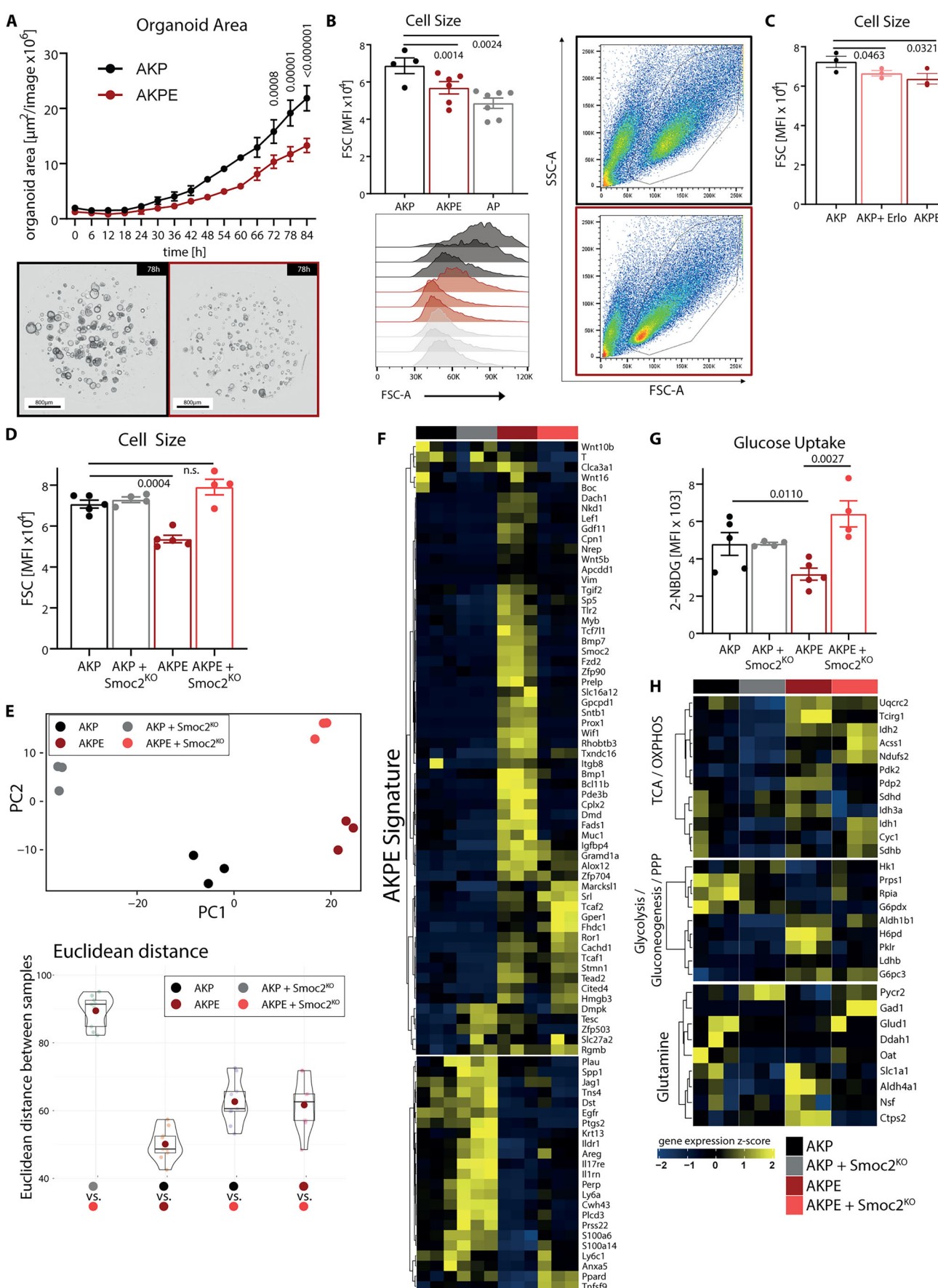

**Figure 5. EGFR deletion alters cell size, while Smoc2 knockout rescues all phenotypes induced by EGFR loss.**

(A) Kinetic quantification of organoid area from AKP and AKPE organoids cultured for 84 h ($n = 3$) and representative images at 78 h. Scale bars, 800 μm. (B) Flow cytometry of mean forward scatter area of AKP, AKPE, and AP organoids ($n = 4$, 6, or 7, respectively). Representative histogram and FSC-A versus SSC-A plots of AKP or AKPE (black or red outlined, respectively). (C) Mean forward scatter area assessed by flow cytometry of AKP, AKP plus 5 μM Erlotinib and AKPE ($n = 3$) and of (D) AKP, AKP + Smoc2[KO] or AKPE and AKPE + Smoc2[KO] organoids ($n = 5$). (E) Principal component analysis (PCA) of RNA-sequencing data of AKP + Smoc2[KO] or AKPE and AKPE + Smoc2[KO] organoids in steady state. Each subpopulation is depicted by indicated color (upper). Euclidean distance between samples of first and second principal component between annotated comparisons, depicting comparison of each sample to all other samples (lower) ($n = 9$). Horizontal lines denote the median and dots the mean. The box limits indicate 25th and 75th percentiles, whiskers extend to 1.5× of the interquartile range (IQR) from the 25th and 75th percentiles. (F) Heatmap showing expression AKPE-signature genes of AKP, AKP + Smoc2[KO], AKPE, and AKPE + Smoc2[KO] organoids in steady state. (G) Flow cytometry analysis ($n = 4$ or $n = 5$, respectively) of mean fluorescence intensity of 2-NBDG of AKP, AKP + Smoc2[KO] or AKPE, and AKPE + Smoc2[KO] organoids. (H) Heatmap showing a subset of metabolic genes of AKP, AKP + Smoc2[KO,] or AKPE and AKPE + Smoc2[KO] organoids. All data represent mean +/− SEM. P-values calculated by paired, two-tailed t-test (between pairs of AKP and AKPE organoids) or one-way ANOVA. Source data are available online for this figure.

proliferation in AKPE organoids (Fig. 7A) suggesting the increased WNT signaling might be required to sensitize CRC to Ras inhibitors.

RNA sequencing analysis of MRTX1133-treated organoids showed that KRAS inhibition in EGFR wild-type (AKP) organoids led to a small increase in the overall AKPE signature score (Fig. 7B). While overall only a small subset of signature genes was transcriptionally changed following Ras inhibition of EGFR wild-type organoids, the analysis allowed to distinguish between genes regulated specifically by Ras, EGFR signaling or both (Fig. EV6A). Genes like *Spp1*, *Wnt16* or *T* were upregulated, while similar to AKPE organoids the expression of the EGFR ligand *Areg* was downregulated following Ras inhibition indicating that its regulation was Ras- and EGFR-dependent (Fig. EV6A). However, upregulation of the WNT and stemness signatures was observed specifically in *Egfr* knockout organoids and especially major target genes like *Smoc2*, *Prox1* or *Lef1* are only altered by EGFR loss, but not by Ras inhibition alone (Figs. 7B and EV6A). Analysis of pathway response genes using PROGENy reaffirmed our previous findings (Fig. 4B) and demonstrates unequivocally that only EGFR loss upregulates WNT response genes, but not KRAS[G12D] inhibition by MRTX1133 (Fig. 7C). As previously reported (Fig. 3A), cell cycle genes were not altered upon EGFR deletion, though treatment with MRTX1133 resulted in a reduction in the expression of cell cycle genes in both AKP and AKPE organoids (Figs. 7D and EV6B). These results highlight the clinical significance of dual inhibition of KRAS and EGFR, as KRAS inhibition alone only modestly reduces proliferation, whereas dual inhibition completely abrogated proliferation.

In summary, our findings emphasize the distinct roles of the EGFR and RAS pathways in CRC development. Specifically, we show that the induction of the WNT- and stemness signature is uniquely driven by EGFR loss and cannot be replicated by KRAS inhibition, suggesting its involvement in contributing to the response to Ras inhibitors.

## WNT and stemness signatures are present in human colorectal tumors

To further validate the findings of our ex vivo AKPE organoid model we next analyzed publicly available single-cell profiled mouse AKP organoids (Data Ref: Qin et al, 2023) (Study 1, Fig. EV7A). As single-cell sequencing does not reliably capture *Egfr* expression, we classified cells according to *Egfr* pathway activity using PROGENy scores (Badia-i-Mompel et al, 2022) (Fig. EV7B). Low *Egfr* pathway activity within KRAS[mt] organoid cells, showed enrichment of our AKPE specific gene-signature (Fig. 8A).

To extend our findings to a human context, we turned to bulk-RNA-Seq of 20 patient-derived organoids (PDOs) (Data Ref: van de Wetering et al, 2015) (Study 2) and primary liver metastasis samples, including matched patient-derived xenografts (PDX) (Data Ref: Leto et al, 2024) (Study 3). Classification into *EGFR* high or low PDO/PDX similarly demonstrated upregulation of our AKPE-derived signature (Fig. 8A). Importantly, treatment of PDX (Study 3) with the anti-EGFR monoclonal antibody cetuximab shows upregulation of 729 genes also upregulated in AKPE organoids, including 60 metabolic genes of which 28 genes are part of the AKPE signature (Fig. 8B). Another, independent study treating PDO with cetuximab showed similar overlapping gene numbers with our AKPE comparison (Study 4, Fig. EV7D) (Data Ref: Herpers et al, 2022).

Furthermore, we validated our EGFR-specific derived gene signature in primary tumor tissues of two recently published single-cell profiled patient cohorts including annotation of KRAS mutation status (Data Ref: Lee et al, 2020; Joanito et al, 2022) (Study 5,6). Focusing our analysis on KRAS[mt] tumor cells (Fig. EV7C), we again observed that cells with downregulated *EGFR* pathway scores show increased AKPE-derived signature scores (Fig. 8C). Comparing DEGs in the Lee et al cohort (Study 4) with our AKPE DEGs demonstrated an overlap of 921 genes (Fig. EV7E).

Testing for clinical significance, we accessed The Cancer Genome Atlas (TCGA) database, containing 456 colorectal cancer patients (COAD, Study 7) (Data Ref: Cancer Genome Atlas Research Network, 2012). Among 165 KRAS[mt] patients, those with low *EGFR* expression (Fig. EV7F) showed enrichment of our AKPE specific gene-signature (Fig. 8C). Comparing DEGs between *EGFR* high and low expressors, revealed 441 differentially expressed genes overlapping with our ex vivo mouse AKPE organoids (Fig. EV7E). Together, across all study modalities and tissue types, stratification of cells or patients into *EGFR* high or low expressors resulted in enrichment of our AKPE-derived gene signature. Strikingly, within the TCGA and an additional independent cohort (Data Ref: Marisa et al, 2013) (Study 7,8), the AKPE signature could only stratify KRAS[mt] patients with AKPE high signature from low signature expressors into better overall survivors, while this was not the case for KRAS[wt] patients (Fig. 8D,E).

Probing other prominent genes of the identified WNT pathway or metabolic genes as e.g., *FZD9*, *WNT6*, or *IDH3A* in patients showed a similar upregulation when EGFR expression was low (Fig. EV7G). Interestingly, by correlating *SMOC2* to *EGFR* expression in patients of the TCGA-COAD cohort, we observed a negative correlation in the subset of KRAS-mutant patients, that was not evident in KRAS[WT] patients (Fig. 8F), corresponding to

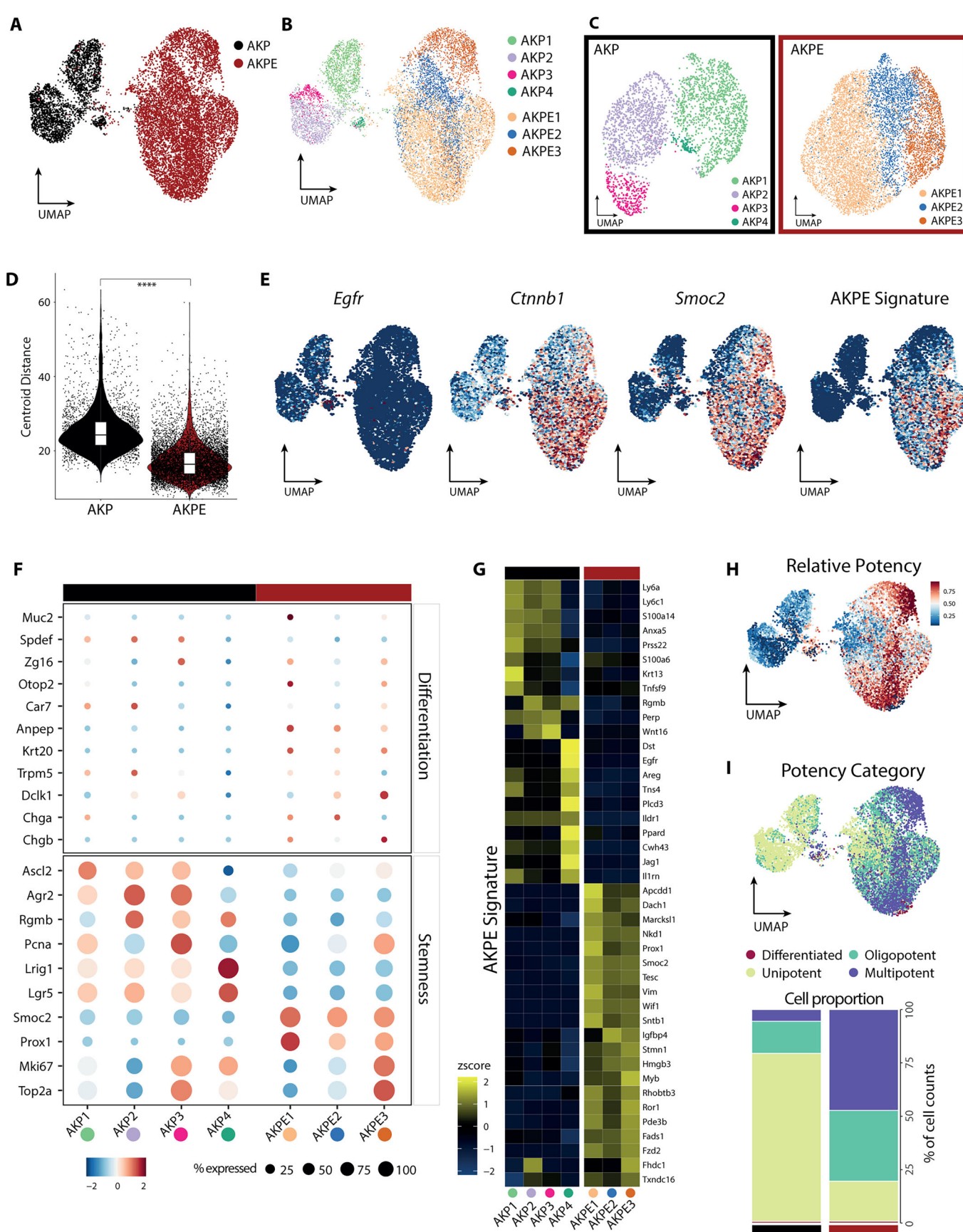

**Figure 6. EGFR deletion reshapes cellular heterogeneity and differentiation trajectories in KRAS^G12D tumor organoids.**

(A) UMAP representation of single-cell transcriptomes of AKP and AKPE organoids colored according to genotype (n = 11,505 cells). (B) UMAP representation of single-cell transcriptomes of AKP or AKPE colored according to identified subcluster. (C) UMAP subcluster representation of single-cell transcriptomes of AKP or AKPE. (D) Violin plot shows heterogeneity assessed by centroid distance of AKP or AKPE cells computed based on principal components with significant variance contribution. Horizontal lines of integrated boxplot depict mean and box limits indicate 25th and 75th percentiles (n = 11,505 cells). (E) UMAP-based visualization of key marker Egfr, Ctnnb1, and Smoc2 expression or AKPE signature score in AKP and AKPE cells. (F) Bubble plot shows relative expression levels of selected differentiation and stemness genes separated by AKP and AKPE subclusters. Color shows expression and dot size represents the percentage of cells expressing the genes in each cluster. (G) Heatmap shows mean expression of AKP signature genes derived from bulk RNA data (Fig. 4) in AKP and AKPE subclusters. (H, I) UMAP representation of single-cell transcriptomes colored by Cytotrace2 relative potency score or potency category. Stacked bar plots indicating proportion of cells categorized by potency (differentiated, unipotent, oligopotent or multipotent). P-values calculated by Wilcoxon Rank-Sum test. Statistical significance: ****P < 0.0001.

previous reports of SMOC2 predicting better clinical outcome in CRC patients (Jang et al, 2020). In line with our mouse organoid results, the negative correlation of *SMOC2* and *EGFR* expression was also seen in all previously assessed validation datasets in bulk or at a single cell level (Fig. EV7H–J). Taken together, we were able to identify a common gene and pathway signature upon deletion of EGFR in KRAS mutated CRC that is reflected in mouse and human datasets.

## Discussion

Precision medicine aims to stratify patients for adequate treatment and KRAS mutations were shown to induce resistance to EGFR inhibition in lung and CRC patients, leading to the exclusion of these patients from targeted anti-EGFR therapies (Eberhard et al, 2005; Pao et al, 2005). As an upstream receptor, EGFR was long thought to be dispensable once downstream constitutive activation of the KRAS oncogene occurs (Van Cutsem et al, 2009; Cox et al, 2014; Brandt et al, 2019). However, several studies challenge this "central dogma" of RAS biology, by demonstrating the necessity to include upstream EGFR inhibition in combination therapies for RAS inhibitors to be effective (Zafra et al, 2020; Zhu et al, 2022; Tria et al, 2023).

In this study, we wanted to understand the specific contribution of EGFR signaling in KRAS mutant tumors by dissecting the effects of EGFR from those of KRAS mutation in CRC. We established a model of murine organoids of advanced CRC and assessed the cell-intrinsic contribution of EGFR to KRAS tumorigenicity. We show that additional deletion of EGFR in APC, TP53, and KRAS^G12D (AKP) triple mutant organoids causes a distinct transcriptional profile with alterations in major metabolic and signaling pathways of MAPK, PI3K, and ErbB. scRNA-seq transcriptomics further uncovered a distinct differentiation pattern upon EGFR loss, resulting in more homogenous cell composition and decreased differentiation status. A unique WNT and stemness gene signature was upregulated in EGFR-deficient CRC organoids and accompanied by a decrease in cell size. Furthermore, we identified the EGFR-specific WNT/CSC signature in a series of human CRC organoid, PDX, PDO and primary tumor sequencing datasets, emphasizing the relevance of our findings for clinical implications.

Critically, we identified *Smoc2* as a prominent gene highly up-regulated upon EGFR deletion. SMOC2 is a secreted extracellular glycoprotein, with increased expression during embryogenesis and is involved in various cellular processes as cell cycle or migration (Jang et al, 2020). Previously, *Smoc2 was* identified as a marker of healthy ISCs and a recent single-cell analysis of primary CRC tumors identified *Smoc2* as a part of a CSC-specific signature, co-

expressed with other CSC markers (Wang et al, 2021). Interaction of the WNT receptors FZD6 and LRP6 with SMOC2 favors receptor-ligand interaction and thereby activates the WNT/β-Catenin signaling pathway (Lu et al, 2019). Most importantly, SMOC2 was described as an independent prognostic marker that correlates with better survival of CRC patients (Jang et al, 2020). In accordance, after EGFR deletion we observe that *Smoc2* exhibits the highest expression within our signature that predicts better overall survival in two independent datasets. We could demonstrate that SMOC2 was the key mechanistic driver of the phenotypic changes observed in CRC organoids lacking EGFR that include metabolic rewiring, reduced glucose uptake, increase in WNT and stemness signatures as well as reduction in cell size. Importantly, additional *Smoc2*-knockout could revert all these phenotypes induced by EGFR deletion, demonstrating that SMOC2 acts as a master-regulator of the identified key pathways in CRC.

Previously, activating mutations of KRAS have been shown to drive increased glucose uptake in CRC tumors, correlating with increased *Glut1* and *Hk2* (GLUT1/hexokinase II) expression (Brown et al, 2018). EGF signaling is generally associated with the Warburg effect in different tumor entities (Hon et al, 2021). In this study, we demonstrate that GLUT1 expression and glucose uptake is reduced in RAS mutant organoids lacking EGFR suggesting that the previous association of increased GLUT1 expression upon KRAS^G12D mutation is at least partially driven by EGFR. SMOC2, which is highly expressed in the absence of EGFR, seems to be a key regulator of this process as we found that additional knock-out of *Smoc2* in EGFR-deficient organoids restores glucose uptake to the levels observed in Ras mutant AKP organoids.

In the intestine, cellular metabolism acts as a central regulator of stem cell fate in order to maintain homeostasis in response to nutrient conditions, injury or insult (Beumer and Clevers, 2021; Urbauer et al, 2021; Li et al, 2022). A recent study demonstrated differential metabolic states in individual cell types of the intestinal niche, where mitochondrial metabolism as evidenced by increased OXPHOS is essential for LGR5^+ ISCs stemness. Simultaneously, differentiated intestinal entities show increased glycolysis (Rodrí-guez-Colman et al, 2017). In the absence of EGFR, we observed a notable shift with glycolytic metabolites increasingly funneled into anaplerotic OXPHOS. This is accompanied by an increase of an entire range of stemness-associated genes, underlining the concept of cancer cells adopting stem-like phenotypes triggered by EGFR deficiency.

WNT is crucial for maintaining physiological stem cell function (Ring et al, 2014; Gehart and Clevers, 2019), particularly in CRC tumors, where tumor-initiating cells rely on aberrant WNT signaling to maintain their stemness and de-differentiated phenotype which is

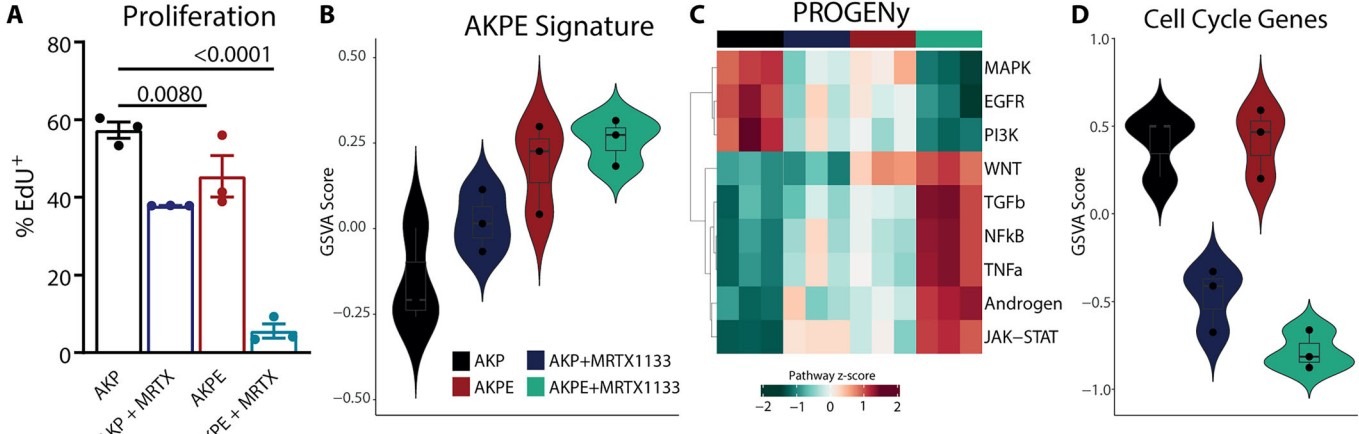

**Figure 7. EGFR deletion implicates synergistic sensitivity to KRAS^G12D inhibition by MRXT1133.**

(A) Flow cytometry analysis ($n = 3$) representing percentage of EdU-positive cells from AKP, AKPE or AKP and AKPE treated with control vehicle or MRTX1133 (100 nM); One-way ANOVA. Data represent mean $+/-$ SEM. (B) Violin plots of GSVA scores for AKPE signature in AKP, AKPE or AKP and AKPE treated with MRTX1133 (100 nM) ($n = 3$). (C) Inference analysis of RNA-seq data with PROGENy. Pathway activity scores are z-scaled and hierarchical clustering was performed by Euclidean distance and complete linkage. (D) Violin plots of GSVA scores for cell cycle genes in AKP, AKPE, or AKP and AKPE treated with MRTX1133 (100 nM) ($n = 3$). (B, D) Horizontal lines denote the median, box limits indicate 25th and 75th percentiles, and whiskers extend from the hinge to the lowest/largest value no further than 1.5x IQR from the 25th and 75th percentiles. Source data are available online for this figure.

observed in 80% of CRC patients carrying inactivating APC mutations (Kleeman and Leedham, 2020). Recent studies implicate an interplay of metabolic identities and WNT signaling, altering stemness and cellular fate (Baulies et al, 2020; Krauß and Gottlieb, 2020). While WNT signaling plays a role in metabolic regulation, we show here that in our experimental system with KRAS mutation, the metabolic phenotype associated with EGFR deletion appears to result from a combination of altered WNT signaling and additional EGFR-dependent pathways. This phenotype can only be partially rescued by either WNT pathway activators or inhibitors.

Recent reports showed that EGFR inhibition in healthy organoids resulted in smaller organoid circumference with high WNT signaling strength (Basak et al, 2017). These results are in line with our finding that upon EGFR deletion or inhibition organoid cell size is smaller and accompanied by increased WNT and stemness signaling, while proliferation is not affected. Importantly, knockout of *Smoc2*, the key mediator of our WNT and stemness signature, rescued the decreased cell size phenotype caused by EGFR deletion.

We demonstrate that proliferation in AKP organoid is not affected by EGFR loss and KRAS inhibition alone through the selective KRAS^G12D inhibitor MRTX1133 results in only modest proliferation reduction. However, MRTX1133 treatment completely blocked the proliferation of AKPE organoids lacking EGFR. Our results are in line with recent pre-clinical studies with MRTX1133 showing synthetic lethality in combination with EGFR inhibition, as cetuximab co-treatment augmented the efficacy of MRTX1133 in in vitro and in vivo PDAC and CRC models (Kataoka et al, 2023; Hallin et al, 2022; Feng et al, 2023). Moreover, in a recent study, heavily pretreated patients with metastatic colorectal cancer were shown to benefit from the treatment of adagrasib (KRAS^G12C inhibitor) in combination with cetuximab (Yaeger et al, 2023). Thus, we propose that the KRAS^G12D allele is critical in maintaining the axis of sustained proliferation, while upstream EGFR performs distinct, non-redundant functions involved

in WNT, stemness and CSC signaling. From our results it is tempting to speculate that increased WNT signaling observed following EGFR inhibition might confer increased sensitivity to RAS inhibitors. Upon EGFR deletion and Ras inhibition in AKP organoids, we also observed that EGFR ligands, such as *Areg*, were downregulated thus interrupting a possible autocrine stimulation loop and providing a possible mechanism why proliferation is affected if both pathways are inhibited. Moreover, impaired EGFR ligand production by tumor cells could also affect CRC development in vivo via paracrine effects on cells of the tumor microenvironment, such as EGFR-positive myeloid cells, which we have previously shown to promote CRC development (Srivatsa et al, 2017). These observations highlight the distinct roles of KRAS and EGFR in regulating CRC growth providing a mechanistic explanation for the clinical outcome of dual targeting of these pathways.

Additionally, our scRNA-seq analysis revealed notable shifts in cellular populations and differentiation status. Both AKP and AKPE organoids lack differentiated cell populations, high potency scores and dispersed expression of stemness genes. Interestingly, the loss of EGFR led to a more homogenous stem cell population with even higher potency scores. suggesting a distinct regulatory role for EGFR in cellular composition and differentiation. Historically, cancer was described as chronic, unidirectional transition of epithelial de-differentiation. However, recent studies have shown that oncogenic mutations collaboratively lock epithelial cells in cancerous states (Qin et al, 2023). Targeting cellular plasticity is evolving as a subject of cancer therapy (Qin and Tape, 2024; Burkhardt et al, 2022). Therefore, our findings shed light on EGFR's role in co-regulating epithelial cell fate trajectory and shaping epithelial heterogeneity and plasticity.

Overall, we demonstrate that EGFR deficiency in KRAS mutant organoids is multifaceted and affects many targets regulating cellular plasticity, metabolism and WNT/CSC activation via increased *Smoc2* expression. These data have important

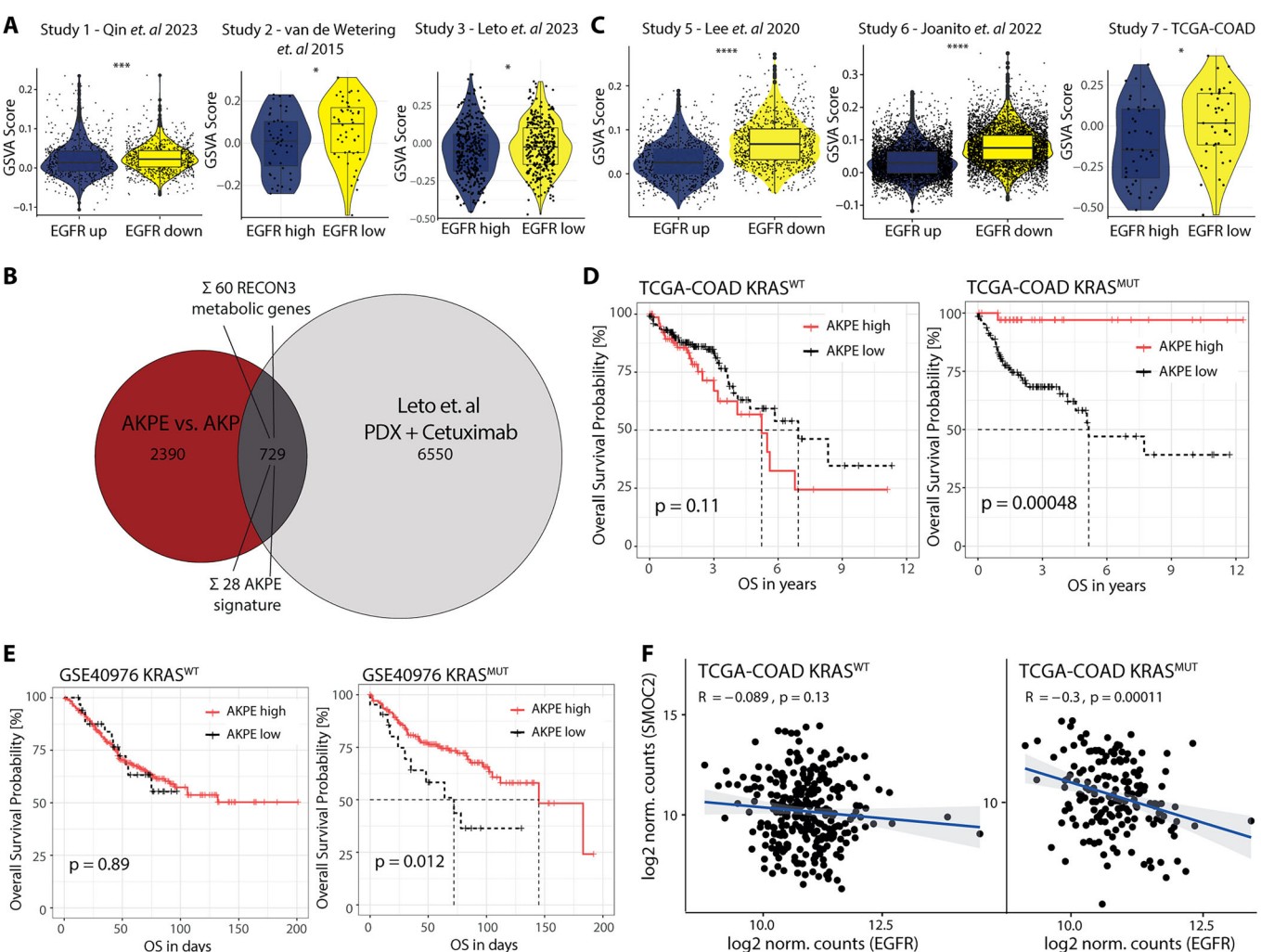

**Figure 8. WNT and stemness signatures are present in human colorectal tumors.**

(A) Violin plots of GSVA scores in cells or patient samples with high or low EGFR expression for indicated publication (Fig. EV7A) (n = 10 organoids, 20 PDXs or 129 PDOs). (B) Venn diagram of mutually regulated genes between AKPE versus AKP and PDX treated with cetuximab versus non-treated from Leto et al (2023). (C) Violin plots of GSVA scores in cells or patient samples with up-or downregulated EGFR pathway activity for indicated publication (Fig. EV7A) (n = 10, 28 or 165 patients). (A, C) Horizontal lines denote the median, box limits indicate 25th and 75th percentiles and whiskers extend from the hinge to the lowest/largest value no further than 1.5x IQR from the 25th and 75th percentiles. (D, E) Kaplan-Meier survival curves of KRAS^wt or KRAS^mt patients comparing AKPE high or low expressors of indicated cohorts. (F) Correlation of EGFR expression with SMOC2 expression in KRAS^wt or KRAS^mt patients in the TCGA-COAD dataset.

translational potential for personalized treatment of CRC as they provide new insights into the current understanding of KRAS biology and resistance mechanisms, emphasizing the significance of incorporating EGFR as a critical player in KRAS mutant tumors.

# Methods

## Reagents and tools table

| Reagent/Resource | Reference or Source | Identifier or Catalog Number |
|---|---|---|
| **Experimental models** | | |
| Mouse: R26Flp; FSF-KrasG12D/+ | (Schönhuber et al, 2014) | N/A |

| Reagent/Resource | Reference or Source | Identifier or Catalog Number |
|---|---|---|
| Mouse: FSF-R26CAG-CreERT2;Trp53lox/lox | (Schönhuber et al, 2014) | N/A |
| Mouse: FSF-R26CAG-CreERT2;R26mT-mG | (Schönhuber et al, 2014) | N/A |
| Mouse: Apc min/+ | | N/A |
| Mouse: Vil/R26creERT2 x EGFRfl/fl | | N/A |
| **Antibodies** | | |
| Anti-Ki67 (1:100) | Abcam | Cat# ab15580, RRID:AB_443209 |
| Glucose Transporter GLUT1 antibody [EPR3915] (1:100,000) | Abcam | Cat# ab115730, RRID:AB_10903230 |
| SMOC2 antibody (1:1000) | Origene | Cat# TA351730 |

| Reagent/Resource | Reference or Source | Identifier or Catalog Number |
|---|---|---|
| HSP90 antibody (1:1000) | Cell Signaling Technology | Cat# 4874, RRID:AB_2121214 |
| beta-Catenin (6B3) Rabbit mAb (1:1000) | Cell Signaling Technology | Cat# 9582, RRID:AB_823447 |
| EGF Receptor (D1P9C) Rabbit mAb (Mouse Preferred) (1:1000) | Cell Signaling Technology | Cat# 71655, RRID:AB_2799807 |
| GSK-3beta (27C10) Rabbit mAb (1:1000) | Cell Signaling Technology | Cat# 9315, RRID:AB_490890 |
| LEF1 (C12A5) Rabbit mAb (1:1000) | Cell Signaling Technology | Cat# 2230, RRID:AB_823558 |
| Phospho-GSK-3 (Ser9) Antibody (1:1000) | Cell Signaling Technology | Cat# 9336, RRID:AB_331405 |
| **Oligonucleotides and other sequence-based reagents** | | |
| Epidermal growth factor receptor (Egfr) fwd: TTGGAATCAATTTTACACCGAAT | Eurofins | N/A |
| Epidermal growth factor receptor (Egfr) rev: GTTCCCACACAGTGACACCA | Eurofins | N/A |
| Lymphoid enhancer binding factor 1 (Lef1) fwd: AGCCGACATCAAGTCATCTTTG | Eurofins | N/A |
| Lymphoid enhancer binding factor 1 (Lef1) rev: TTCTCTGGCCTTGTCGTGG | Eurofins | N/A |
| SPARC related modular calcium binding 2 (Smoc2) fwd: GCTTGGGTGTCACCAGAGAG | Eurofins | N/A |
| SPARC related modular calcium binding 2 (Smoc2) rev: TGGGCTGTCTATTAGAAGAAGAAC | Eurofins | N/A |
| TATA-binding protein (Tbp) fwd: GGGGAGCTGTGATGTGAAGT | Eurofins | N/A |
| TATA-binding protein (Tbp) rev: CCAGGAAATAATTCTGGCTCAT | Eurofins | N/A |
| SMOC2: Hs00405777_m1, FAM-MGB | Fischer Scienific Austria | 4448892 |
| GAPDH: Hs02758991_g1, VIC-MGB | Fischer Scienific Austria | 4448484 |
| **Chemicals, Enzymes and other reagents** | | |
| Erlotinib, Free Base | Santa Cruz Biotechnology | Cat# sc-396113 |
| DMSO | Sigma-Aldrich | Cat# D8418 |
| **Critical Commercial Assays** | | |
| SYTOX™ Blue Dead Cell Stain | Thermo Fisher Scientific | Cat#10297242 |
| miRNeasy Micro Kit | Qiagen | Cat# 217084 |
| Power SYBR® Green PCR Master Mix | Thermo Fisher Scientific | Cat# 10209284 |
| Mito Stress Test assay | Agilent | Cat# 103010-100 |
| Chromium Next GEM Single Cell 5' Kit v2 | 10x Genomics | Cat# 1000265 |
| ClickTech EdU Cell Proliferation kit | BaseClick | Cat# BCK-EdU488FC50 |
| **Software** | | |
| FlowJo v10 | FlowJo, LLC | https://www.flowjo.com/ |
| GraphPad Prism 5 | GraphPad Software | https://www.graphpad.com/ |
| Seahorse Wave Pro | Agilent | https://www.agilent.com/ |
| IncuCyte S3 Live® Cell Analysis System | Sartorius | https://www.sartorius.com/ |

| Reagent/Resource | Reference or Source | Identifier or Catalog Number |
|---|---|---|
| Adobe Illustrator CS6 | Adobe Inc. | https://www.adobe.com/ |
| TissueFAXs | TissueGnostics GmbH | https://tissuegnostics.com/ |
| TraceFinder 4.1 | Thermo Fisher | https://www.thermofisher.com/ |
| Halo v5.0 | Indica Labs | https://indicalab.com/halo/ |
| R 4.4.2 | The Comprehensive R Archive Network | https://cran.r-project.org/ |
| CellSens | Evident Corporation | https://evidentscientific.com/ |
| **Other** | | |
| RNA-seq dataset | This study | GSE263594 |
| RNA-seq dataset | This study | GSE289455 |
| RNA-seq dataset | This study | GSE289571 |
| RNA-seq dataset | This study | GSE289570 |
| RNA-seq dataset | This study | GSE289498 |
| Single-Cell Seq dataset | This study | GSE289737 |
| Single-Cell Seq dataset | Qin et al, 2023 | PRJNA88361 |
| RNA-seq dataset | van de Wetering et al, 2015 | GSE65253 |
| RNA-seq dataset | Leto et al, 2024 | GSE204805 |
| RNA-seq dataset | Herpers et al, 2022 | GSE186531 |
| Single-Cell Seq dataset | Lee et al, 2020 | GSE132465, GSE132257 and GSE144735. |
| Single-Cell Seq dataset | Joanito et al, 2022 | syn26720761, syn26844071 |
| RNA-seq dataset | TCGA-COAD | https://portal.gdc.cancer.gov/projects/TCGA-COAD |
| MicroArray dataset | Marisa et al, 2013 | GSE40967 |

## Mice

To model human CRC reflecting the sequential mutations occurring during multi-step tumor development, we employed a complex breeding scheme using FSF-Kras$^{G12D}$ mice (kindly provided by Dieter Saur, TU Munich, Germany) that rely on the dual recombination system based on Cre/loxP and Flp/frt (Schönhuber et al, 2014). Oncogenic K-Ras$^{G12D}$ is activated in intestinal epithelial cells by the action of Villin-Flp (provided by D. Saur) in an Apc$^{min}$ EGFR$^{f/f}$ background. Vil-Flp-mediated recombination also leads to TP53 tumor suppressor inactivation of the *FRT*-flanked *Trp53* alleles (*Trp53$^{frt}$*, kindly provided by D. Saur). An additional dual-reporter gene, in which firefly luciferase and eGFP are activated upon Flp recombination and renilla luciferase and tdTom following Cre recombination (R26$^{dual}$), allows monitoring of tumor development by in vivo imaging (Schönhuber et al, 2014). Mice heterozygous for the germ line truncation allele *Apc$^{min/+}$* are highly susceptible to spontaneous loss of the second allele leading to APC-deficient intestinal epithelial cells and adenoma formation. For inducible deletion of *Egfr*, we crossed *Egfr$^{fl/fl}$* mice to mice harboring FSF-silenced-CreER$^{T2}$ under the Rosa26 or Villin promoter (FSF-R26$^{CAG-CreERT2/+}$/FSF-Vil $^{CreERT2/+}$;*Egfr$^{fl/fl}$*). Simplified,

gene deletion or activation was designated as A ($APC^{min/+}$), K ($KRAS^{G12D/+}$), P ($Trp53^{frt}$), or E ($EGFR^{fl/fl}$), resulting in AKP, AKPE, and AP combinations, respectively. Female and male mice (age between 8 and 16 weeks) of C57/BL6 genetic background were used for breeding. Genotyping was performed as previously described (Schönhuber et al, 2014; Srivatsa et al, 2017).

All animal experiments were compliant with federal laws and guidelines of the Medical University of Vienna. Animal experimental procedures were approved by the Animal Experimental Ethics Committee of the Medical University of Vienna and the Austrian Federal Ministry of Science and Research (animal license numbers: BMBWF-66.009/0319-V/3b/2019). Mice were housed in the animal facility of the Medical University of Vienna at constant room temperature, unlimited access to water and standard laboratory chow and 12 h light-dark cycle. All mice were closely monitored by the authors, animal care technicians, and by a veterinary scientist responsible for animal welfare.

## Mouse tumor organoid isolation and cultivation

Colons containing tumors were dissected from male and female mice of the respective genotypes, flushed with cold PBS, opened longitudinally and polyps collected and roughly chopped with scissors. Dissected material was enzymatically digested with 2 mg/mL collagenase I (Stemcell Technologies, Ord. Nr. 07415) in Advanced DMEM/F12 (Thermo Scientific, Cat#12634028) for 20 min at 37 °C with vigorous pipetting every 5 to 10 min. Digested tissue fragments were investigated under the microscope to determine if sufficient dissociation of crypts occurred, then washed with Adv. DMEM/F12 with 20% FBS (Sigma) and passed through a 70 μm cell strainer. Low centrifugation forces (100 G) and decreased deacceleration enabled the collection of small tissue fragments containing crypts and to keep single cells in suspension. Isolated tissue fragments were plated in cold Matrigel (Growth Factor Reduced, Corning 356231) and seeded as 5–10 μL droplets into cell culture plates. After 5–10 min incubation at 37 °C upside down, mouse tumor organoids were overlayed with a prewarmed culture medium composed of Advanced DMEM/F12 (Thermo Scientific, Cat#12634028) supplemented with 50% RSpondin, Noggin and Wnt3A conditioned medium, 10 mM HEPES (Thermo Scientific, Cat#15630080), 2 mM GlutaMAX (Thermo Scientific, Cat#3550061), 1% Penicillin-Streptomycin solution, B-27 without RA (Thermo Scientific, Cat#17504044), 1 mM N-Acetylcystein (Sigma, A7250), 50 ng/mL murine Egf (Peprotech), 10 μM Y27632 (STEMCELL Cat #72304) and 1 μM galunisertib (THP LY2157299). Organoid clones were tested monthly for mycoplasma contamination by PCR. The following organoid clones were employed in this study: AP: AP1 (male, clone 5510), AP2 (female, clone 5593), AP3 (male, clone 5596); AKP: AKP1 (male, clone 3806), AKP2 (male, clone 3773), AKP3 (female, clone 3934); AKP1-3 were treated with Tamoxifen or Adeno-Cre to delete the EGFR resulting in the paired AKPE1-3 organoids.

## Cultivation and assay conditions/inhibitor assays/treatment

To passage organoids, Matrigel drops were mechanically disaggregated and treated with Trypsin-EDTA (Thermo Scientific Cat#25200056) for 20–40 min at 37 °C to achieve single-cell suspension. Cells were washed with advanced DMEM/F12 with 20% FCS and spun at 100 G for 5 min. Pelleted cells were seeded at a density of 10E4/100 μL Matrigel, overlayed with culture medium and incubated at 37 °C with 5% $CO_2$. To assess the doubling rate, cells were passaged for a minimum of four consecutive passages. At each passage, cells were collected using 0.05% trypsin-EDTA for 20–40 min until single-cell suspensions were achieved. Cells were manually counted with 1:1 diluted trypan blue to assess cell viability with a hemocytometer. After counting, cells were seeded at a density of 10E4/100 μL Matrigel into one well of a 6-well plate. For assessment of doubling rate, final cell number over initial seeded cell number was calculated and averaged over the total passage number for each organoid line. For EdU incorporation, organoids were cultured for 48 h, Ethynyl-deoxyuridine was added at a final concentration of 10 μM and incubated for 2 h. Organoids were trypsinized to single-cell suspension and the ClickTech EdU Cell Proliferation kit (BCK-EdU488FC50) was used according to manufacturer's description. Cells were counterstained with PI and analyzed with flow cytometry. For inhibitor experiments, compounds were added at the following concentrations: 10 μM BPTES (Sigma, SML0601), 2 μM ICG-001 (Sigma, 5047120001), 3 μM CHIR99021 (Sigma, SML1046), 100 nM MRTX1133 (MedChemExpress, HY-134813), 5 μg/mL cetuximab (Merck) for 48 h.

## IncuCyte assay

Organoids were trypsinized to single cells and seeded in quadruplicates at a density of 1000 cells/well in a 24-well plate. Organoid size, proliferation and were monitored and analyzed with the IncuCyte S3 Live® Cell Analysis System every 6 h for up to 6 days.

## Flow cytometry

Matrigel drops containing organoids were mechanically disaggregated and incubated with Trypsin-EDTA (Thermo Scientific Cat#25200056) for at least 40 min at 37 °C to achieve single-cell suspension. Cells were washed with advanced DMEM/F12 with 20% FCS, filtered through a 40 μm filter and used for subsequent FACS analysis. Cell suspensions were stained with Ki67 antibody (Abcam Cat#ab15580). Intracellular FACS stains were performed as previously described (Novoszel et al, 2021). Shortly, 10E6 cells were permeabilized with BD Cytofix/Cytoperm Kit (BD Biosciences), stained with respective antibodies and recorded using BD Fortessa X10 machine. FITC Annexin V/7-AAD was examined using the Annexin A5 Apoptosis Detection Kit (BioLegend) or APC Annexin V (Biozym Cat# 640920). FACS analysis was performed by FlowJo V10 software.

## Western blot

Before cell lysis, organoids were liberated from Matrigel using Cell Recovery Solution (Corning, 354253). Western blot analysis was performed as previously described (Linder et al, 2018). The complete collection of antibodies used is listed in the Reagents and tools table. Briefly, cell pellets were lysed with RIPA buffer containing Protease Inhibitor (Roche), PMSF, Sodium orthovanadate and sodium fluoride. Denatured proteins were resolved in SDS-Polyacrylamide gels and transferred to nitrocellulose

membranes, which were subsequently blocked with TBST (Tris-buffered saline with tween20) + 5% BSA. Primary antibodies were incubated overnight at 4 °C, secondary HRP-conjugate antibodies were used at 1:5000 for 1 h at room temperature. ECL-enhanced chemiluminescence (Bio-Rad, Ca, USA) and ChemiDoc (Bio-Rad, CA, USA) were used for development.

## Fluorescence microscopy and expression level analysis

Organoids were liberated by incubating 250 μL of CellRecovery solution (Corning, 354253) for 20 min on ice and embedded in OCT compound (Tissue Tek). Frozen sections were processed as previously described (Strobl et al, 2024) and 1:100 primary antibody dilution were used for staining. Total expression of immunofluorescence-stained organoid sections have been scanned with the Olympus VS200 slide scanner (Evident Corporation) with fixed illumination and camera exposure settings and a 40 × 0.95 na magnification air immersion objective lens. Images were acquired in the.vsi file format. Total expression levels where quantified in the HALO 4.0 (Indica Labs) software using the Highplex FL package and HALO AI for Cell segmentation based on the DAPI stained nuclei. Background and staining threshold have been adjusted with a secondary only control sample, measured at same setting.

Nuclear vs. membrane levels of immunofluorescence-stained organoid sections have been scanned in 40 × 0.95 na air magnification with the Olympus VS200 slide scanner (Evident Scientific) with fixed illumination and camera exposure settings. Images were acquired in 16 bit in the (.vsi) file format. Auto fluorescence and thresholds have been adjusted with correlating secondary only samples, acquired at same settings. The CellSens imaging software (Evident Corporation) and the Count and measure module has been used to quantify signal intensity values in nuclei and membranes. Cell, nuclear and membrane segmentation was achieved by the deep learning package and training neuronal networks for the detection of nuclei, based on DAPI as well as membranes based on a WGA (Wheat Germ Agglutinin) (Thermo-Fisher Scientific).

## Sample preparation and metabolite extraction for metabolomics stable isotope tracing

For each experiment, dissociated organoids were seeded in four to six replicates at a density of 8.75E04 cells per 25 μL Matrigel into a well of a 24-well plate. Organoids were grown in complete medium for 24 h, then washed twice with PBS and growth medium was replaced with metabolic medium for equilibration: DMEM-F12 no glutamine (Thermo Scientific, Cat# 21331046) or custom order of DMEM-F12 no glucose (Biological Industries), 10 mM HEPES (Thermo Scientific, Cat#15630080), 400 mg/mL BSA (Capricorn-scientific), 1 mM N-Acetylcystein (Sigma, A7250), 50 ng/mL murine Egf (Peprotech), 100 ng/mL murine noggin (STEMCELL), 1 μM galunisertib (THP LY2157299), 2 mM glutamine (Sigma, G7513), ITS-X (1:100, ThermoFisher Scientific, 51500056), 1 mg/mL glutathione (Sigma-Aldrich, G6013), 0.3 ng/mL ammonium metavanadate (Sigma-Aldrich, 204846), 0.25 nM manganese chloride (Sigma-Aldrich, 244589) and 2.5 mg/L ascorbic acid (Sigma-Aldrich, A8960). After 12 h, medium was replaced with fresh metabolomics medium supplemented with either unlabeled

glutamine/glucose or 200 mM U-$^{13}$C$_5$ glutamine (Sigma-Aldrich Cat#605166)/10 mM U-$^{13}$C$_6$ glucose (Sigma), respectively. 15 min before the endpoint for glucose or 30 min for glutamine stable isotope tracing, replicates with medium of unlabeled wells were changed to metabolomics medium with labeled metabolite. Wells, containing only medium was taken as control. At endpoint, extracellular medium/supernatant metabolites were extracted by 1:50 mixing with cold extraction mix consisting of LC-MS grade methanol (#1060351000, Sigma-Aldrich), acetonitrile (#1000291000, Sigma-Aldrich) and milliQ-water in a proportion of 5:3:2. Matrigel domes were washed once with PBS and organoids were liberated by incubating 250 μL of CellRecovery solution (Corning, 354253) for 20 min on ice. Cellular metabolites were extracted by adding 90 μL of extraction mix to cell pellets. Samples were vortexed and immediately centrifuged at maximum speed for 10 min at 4 °C. Supernatants were transferred, kept at −80 °C until LC-MS analysis.

## Targeted metabolomics

LC-MS isotope tracing analysis was performed as previously described (Mackay et al, 2015). In short, the Thermo Ultimate 3000 high-performance liquid chromatography (HPLC) system linked to Q-Exactive Orbitrap Mass Spectrometer (Thermo Fisher Scientific) was operated at a resolution of 35,000 at 200 mass/charge ratio (m/z), with electrospray ionization and polarity switching mode to allow both positive and negative ions across mass ranges of 67 to 1000 m/z. HPLC setup containing a ZIC-pHILIC column (SeQuant; 150 mm × 2.1 mm, 5 μm; Merck) and a ZIC-pHILIC guard column (SeQuant; 20 mm × 2.1 mm). In total 5 μL of the extracted metabolites were injected and compounds were separated by a mobile phase gradient of 15 min, starting at 20% aqueous (20 mM ammonium carbonate adjusted to pH 0.2 with 0.1% of 25% ammonium hydroxide) and 80% organic (acetonitrile) and finished with 20% acetonitrile. During the total run time of 27 min, flow rate and column temperature were set at 0.2 mL/min and 45 °C. All metabolites were detected using mass accuracy below 5 ppm. Thermo Xcalibur was used for the data acquisition.

## Metabolic data analysis

Collected data were analyzed with TraceFinder 4.1 (Thermo Fisher Scientific) by determining peak areas of metabolites using an exact mass of singly charged ions. In-house mass spectrometry metabolites library was built by running commercially available standards to predetermined retention times of metabolites on the pHILIC column. Protein debris was used to quantify total protein concentration with Pierce Bradford BCA and used for normalization. Metabolite Autoplotter 2.5 was used for data visualization (Pietzke and Vazquez, 2020).

## Metabolite absolute quantification

For glucose, glutamate, lactate and glutamine absolute quantification, calibration standards of 17.5 mM glucose, 0.05 mM glutamate, 2 mM glutamine, and 10 mM lactate were prepared and serially diluted. Calibration curves were prepared in a metabolic tracing medium and identically extracted as culture samples.

## Seahorse metabolic assay

Organoids were dissociated to single cells and seeded at a density of 1300 cells/2 μL Matrigel per well into a 96-well Seahorse plate with complete organoid medium 48 h prior to assay. Matrigel domes were washed twice with PBS for 10 min and pre-warmed 180 μL Seahorse XF media (Agilent, Les Uli, France) containing 10 mM glucose, 2 mM glutamine and 1 mM pyruvate per well were added and Mito Stress Test assay was performed as described by the manufacturer from this point. Optimized compound concentrations of 1.5 μM oligomycin, 2 μM FCCP, and 0.5 μM antimycin/rotenone were used. The OCR and ECAR values were normalized to organoid confluence percentage with IncuCyte S3 Live® Cell Analysis System.

## RNA extraction and RT–qPCR analysis

Total RNA was isolated from organoids cultured 48 h after seeding using RNAeasy isolation kit following manufacturer instruction (Qiagen). cDNA was synthesized with ProtoScript II Reverse Transcriptase (NEB) as described by manufacturer's instructions. For real-time quantitative PCR (qRT-PCR) Power SYBR Green Master Mix (Thermo Fisher Scientific, Cat# 10209284) and CFX96 Real-Time System (Bio-Rad) were used. Primers used for qRT-PCR are listed Reagents and tools table. Expression data were calculated using the deltaCt methods and normalization was achieved in relation to tbp mRNA levels.

## Single-cell enzyme activity assays

Single-cell enzyme activity assays for GAPDH and G6PDH were based on previously reported enzymehistochemistry methods (Miller et al, 2017; Noorden and Frederiks, 1993). We optimized assay conditions to assess the mean enzyme activity per cell of AKPE and AKP organoids, which were embedded in the optimal cutting temperature (O.C.T.) medium (Sakura Finetek, USA) and snap-frozen. Briefly, samples were sectioned into 7 μm slices on glass slides and stored at −80 °C for at least overnight. Activity assays were performed by thawing slides for 3 min at RT and then incubating sections with an enzyme specific reaction mixture in 68 mM Tris-Maleate buffer at pH 8.5 for GAPDH or pH 7.5 for G6PDH containing: 10% polyvinyl alcohol, 0.45 mM methoxyphenanzine methosulfate, 5 mM sodium azide, 2 mM nitroblue tetrazolium chloride and for GAPDH assay 2.5 mM glyceraldehyde 3-phosphate and 3 mM NAD +, while for G6PDH assay 15 mM glucose-6-phosphate, 0.8 mM NADP+ and 4 mM $MgCl_2$ was added. Incubation time for GAPDH activity assays was 15 min and 10 min for G6PDH assays. To ensure specificity of the signals, we additionally included enzyme inhibitors iodoacetate (167 mM) for GAPDH and dehydroepiandrosterone (8.3 mM) for G6PDH in the reaction mixtures as negative control. The activity assays were stopped by washing with pre-warmed (60 °C) PBS and sildes were subsequently stained with DAPI, and mounted with Fluoromount-G. Images were acquired in 8-bit using a fluorescent camera and a 20x objective on Zeiss Axio Imager TG-OEM and TissueFAXs acquisition software (TissueGnostics GmbH). The acquired images of the enzyme activity assays were subsequently analyzed using StrataQuest (TissueGnostics GmbH). Briefly, transmission images of the enzymehistochemistry staining were inverted and cell segmentation was performed based on DAPI staining. The obtained mean intensity values were reported as mean enzyme activity per cell. For visual representation of the obtained activity assays we used a 20x objective and bright-field camera on PhenoImager (Vectra Polaris, Akoya Biosciences).

## Bulk RNA-sequencing and data analysis

Library preparation and total transcriptome full-length mRNA sequencing was performed by Novogene UK using the NovaSeq 6000 Illumina Platform. Preprocessing and mapping of sequence reads to the mouse genome GRCm38/mm10 and GENECODE M25 annotations was achieved by the nf-core RNASeq pipeline version 3.2 (Ewels et al, 2020). Briefly, reads were mapped by STAR 2.6.1 d (Dobin et al, 2013) and quantified by Salmon v1.4.0 (Patro et al, 2017). Subsequent analysis was performed using the statistical computing environment R v. 4.1.0 (R Core Team, 2022). Differential gene expression levels were identified by Bioconductor package DESeq2 (Love et al, 2014) and interpretation of biological significance of differential gene expression, Gene Ontology enrichment analysis was performed with the package ClusterProfiler v4.1.0 (Yu et al, 2012). GSEA analysis was performed for gene lists using signal-to-noise ratios for gene ranking and significance of enrichment was estimated using 1000 gene permutation. Pathway activity scores were assessed from normalized counts by the PROGENy (Pathway RespOnsive GENes) approach (Schubert et al, 2018). Heatmaps generated using ComplexHeatmap v. 2.10.0 (Gu et al, 2016) and ggplot2 (Wickham, 2016). Metabolic subsetting RECON3D (Brunk et al, 2018; Khodaee et al, 2020). Total lists of statistically significant differentially expressed genes identified are provided in Dataset EV2.

## Single-cell RNA-sequencing and data analysis

AKP and AKPE organoids were seeded at 10E4/100 μL Matrigel for 48 h, trypsinized, and to single cell suspension were subjected to scRNA-seq using the 10x Genomics Chromium Next GEM Single Cell 3' Reagent Kit v3.1 (Dual Index) (10X Genomics PN-1000269) and sequenced with the NovaSeq X Plus System (PE150-10B). scRNA-seq data processing raw binary base call (BCL) sequence files were converted to FASTQ files and processed with the 10X Genomics Cell Ranger pipeline version-8.0.1 using the GRCm39-2024-A reference genome. ScRNA-seq data analysis was performed using *Seurat* (version 5.1.0) (Hao et al, 2021). The analysis pipeline encompasses quality control, data normalization, dimensionality reduction, cell clustering and visualization (Hua et al, 2024). Quality control removed genes found in less than 5 cells and cells with <200 transcripts, >1000 unique molecules, >20% mitochondrial gene ratio and doublets identified by *scDblFinder* (Germain et al, 2022) resulting in 2915 AKP and 8590 AKPE high quality cells. Normalization was performed using the *SCTransform* function, with percentage of mitochondrial reads as variables to regress out. Cell clustering was computed using the *FindNeighbours* and *FindClusters* Seurat functions with first 30 principal component and a resolution of 0.2.

To determine the heterogeneity of organoids, we calculated the centroids of all principal components, which significantly contributed to the variance, and subtracted them from the original PCA embedding, so that the centroid was repositioned at the origin

**The paper explained**

**Problem**

Epidermal Growth Factor Receptor plays an important role in colorectal cancer, but its inhibition has been considered ineffective in KRAS mutant tumors. Recent evidence suggests that co-targeting EGFR and RAS may be a promising therapeutic strategy, but the tumor-intrinsic role of EGFR in KRAS mutant colorectal cancer remains unclear.

**Results**

Our study used mouse colorectal cancer organoids carrying KRAS, APC, and TP53 mutations, allowing for inducible EGFR deletion to investigate its role in KRAS mutant organoids. Metabolomic analysis revealed significant changes, with a shift away from glycolysis and an increase in glutaminolysis upon EGFR deletion in isolated organoids. Loss of EGFR triggered a well-defined transcriptional response, leading to the downregulation of key signaling pathways, including MAPK, PI3K, and ErbB. Additionally, EGFR deletion activated a distinct cancer stem cell (CSC)/WNT signature accompanied by reduced cell size. Mechanistically we identified SMOC2 as the key target upregulated upon EGFR loss driving the activation of the WNT and stemness pathway, the reduction in cell size as well as the metabolic changes. Finally, by analyzing patient datasets, we demonstrated that the identified transcriptional signature is associated with improved overall survival in RAS mutant CRC patients, highlighting its potential clinical relevance.

**Impact**

This study provides a mechanistic explanation of how EGFR signaling impacts KRAS mutant colorectal cancer, challenging the notion that EGFR inhibition is ineffective in these tumors. By identifying a transcriptional signature linked to patient survival, the findings could help refine therapeutic strategies and patient stratification. The results also support the emerging idea that co-inhibiting EGFR and RAS could be a viable treatment approach for KRAS mutant CRC, potentially improving patient outcomes.

position. Then, the distances from the centroid to all cells were computed, with each principal component scaled by its corresponding standard deviation. The distances were then summarized by their mean. CytoTRACE2 (version 1.0.0) (Kang et al, 2024) with default parameters was utilized to predict differentiation states and scores.

### Human mRNA expression data

Publicly available human colon adenocarcinoma (COAD) mRNA expression data from The Cancer Genome Atlas (Cancer Genome Atlas Research Network, 2012) were accessed via the GDC Data Portal (https://portal.gdc.cancer.gov/projects/TCGA-COAD). Corresponding mutation and clinical data were downloaded through the Broad Institute GDAC firehose website (https://gdac.broadinstitute.org/).

scRNA sequencing data of the SMC cohort, as reported by Lee et al, 2020 (Lee et al, 2020) were accessed from the NCBI Gene Expression Omnibus database (Barrett et al, 2013) under the accession number GSE132465. Provided raw counts matrix was filtered, normalized using the most recent version of Seurat functions (Seurat V4.1.1) (Hao et al, 2021) and applying filtering criteria described by the authors. Six major cell type annotations and metadata as provided, were cross-checked for SMC cohort subset and utilized for further analysis. We performed single-cell pathway analysis using PROGENy integrated into decoupleR (Badia-i-Mompel et al, 2023) with top 1000 target genes of the progeny model, as recommended for single-cell data. Differential gene expression analysis was estimated through default Wilcoxon Rank-Sum test of the Seurat

'FindMarkers' function. P value adjustments for multiple testing were corrected using Bonferroni correction based on total number of genes in the respective comparison. All processing was performed in R version 4.4.2 (R Core Team, 2022).

### Statistics

Comparison of two matched groups was calculated with paired or unpaired Student's t-test, as applicable. All analyses were calculated using GraphPad Prism 5 software. Significant differences between experimental groups were stated as: $*p < 0.05$, $**p < 0.01$, $***p < 0.001$, or $****p < 0.0001$. The data are shown as mean ± SEM. All experiments were conducted non-blinded, and we included all samples in our analyses.

## Data availability

Raw sequencing data for this study are available at NCBI GEO under the accession number GSE263594 (bulk RNA-seq), GSE289455 (SMOC2 RNA-Seq), GSE289571 (WNT-Inh), GSE289570 (BPTES), GSE289498 (MRTX1133-Inhibition), GSE289737 (single cell RNA-seq).

The source data of this paper are collected in the following database record: biostudies:S-SCDT-10_1038-S44321-025-00240-4.

## Peer review information

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

## Acknowledgements

We would like to thank Dieter Saur for providing some of the mouse strains used in this study (FSF-R26 dual-reporter, FSF-p53, FSF-Kras^G12D; Villin-Flp); Martina Hammer and the staff of the Department of Biomedical Research of the Medical University of Vienna for maintenance of the mouse colonies, Thomas Bauer for designing the graphical abstract, Christine Tuppy and Luise Bellach for help with genotyping; Johannes Reisecker for assistance in flow cytometry experiments. Alexandra Bogusch and Temenuschka Baykuscheva-Gentscheva for technical assistance. Luke Zappia for making data analysis and presentation highly efficient (Zappia). This work was funded by the following grants acquired by MS: the European Union's Horizon 2020 research and innovation program under the Marie Skłodowska-Curie grant agreement No 766214 (META-CAN), the Vienna Science and Technology Fund (WWTF) grant LS16-025, the European Research Council (ERC) Advanced grant (ERC-2015 AdG TNT-Tumors 694883), the Austrian Science Fund FWF, DK-PhD program W1212 "Inflammation and Immunity", DocFunds DOC 32-B28 "Tissue Home" and DOC 59-B33 "IPPTO". This work was also supported by grants acquired by DK: the Fellinger Cancer Research Fund and the City of Vienna Fund for Innovative Interdisciplinary Cancer Research. AH and NB were supported by FWF Sonderforschungsbereich F83.

## Author contributions

**Dana Krauß**: Conceptualization; Data curation; Formal analysis; Funding acquisition; Validation; Investigation; Visualization; Methodology; Writing—original draft; Project administration; Writing—review and editing. **Veronica Moreno-Viedma**: Resources; Data curation; Formal analysis; Methodology; Writing—review and editing. **Emi Adachi-Fernandez**: Resources; Data curation; Formal analysis; Methodology; Writing—review and editing. **Cristiano De Sá Fernandes**: Data curation; Formal analysis; Writing—review and editing. **Jakob-Wendelin Genger**: Data curation; Formal analysis; Writing—review and editing. **Ourania Fari**: Data curation; Formal analysis; Methodology; Writing—review and editing. **Bernadette Blauensteiner**: Data curation; Formal analysis; Writing—review and editing. **Dominik Kirchhofer**: Data curation; Formal analysis; Writing—review and editing. **Nikolina Bradaric**: Data curation; Formal analysis. **Valeriya Gushchina**: Methodology. **Georgios Fotakis**: Resources; Formal analysis; Supervision; Writing—original draft; Writing—review and editing. **Thomas Mohr**: Resources; Writing—review and editing. **Ifat Abramovich**: Resources; Data curation; Formal analysis; Supervision; Investigation; Visualization; Methodology; Writing—review and editing. **Inbal Mor**: Conceptualization; Data curation; Supervision; Investigation; Methodology; Writing—review and editing. **Martin Holcmann**: Data curation; Formal analysis; Methodology; Writing—review and editing. **Andreas Bergthaler**: Resources; Writing—review and editing. **Arvand Haschemi**: Resources; Formal analysis; Supervision; Validation; Visualization; Writing—review and editing. **Zlatko Trajanoski**: Resources; Supervision; Writing—review and editing. **Juliane Winkler**: Resources; Data curation; Methodology; Writing—review and editing. **Eyal Gottlieb**: Conceptualization; Resources; Supervision; Funding acquisition; Investigation; Methodology; Writing—original draft; Writing—review and

editing. **Maria Sibilia**: Conceptualization; Resources; Formal analysis; Supervision; Funding acquisition; Validation; Visualization; Writing—original draft; Project administration; Writing—review and editing.

Source data underlying figure panels in this paper may have individual authorship assigned. Where available, figure panel/source data authorship is listed in the following database record: biostudies:S-SCDT-10_1038-S44321-025-00240-4.

## Disclosure and competing interests statement

The authors declare no competing interests.

# Expanded View Figures

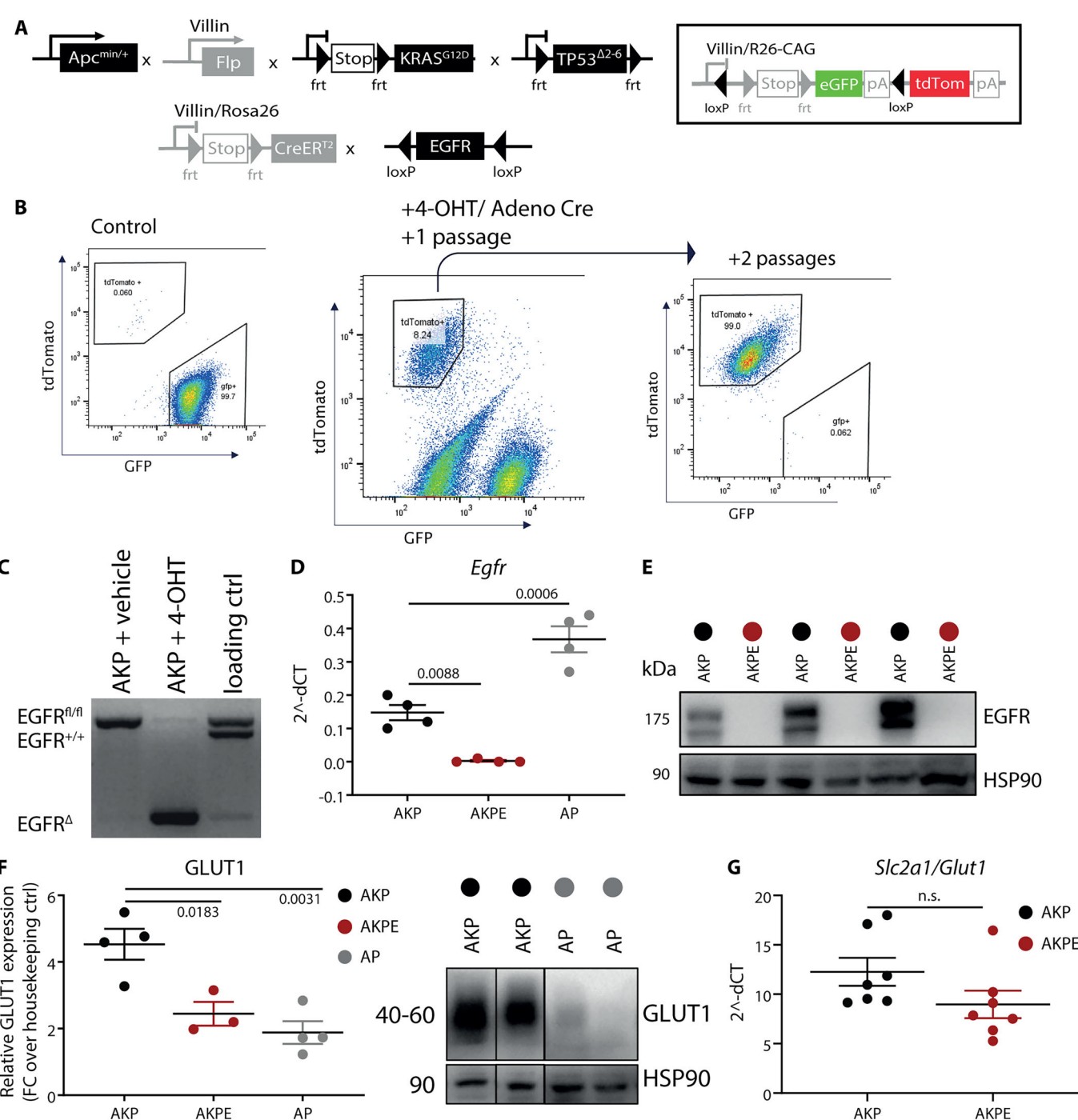

**Figure EV1.  Establishment and molecular characterization of EGFR-deficient organoids isolated from GEMMs.**

(A) Schematic depiction of genetic crossing strategy used for generating GEMMs, from which organoids were derived. (B) Representative FACS gating strategy of AKP (green fluorescent protein (GFP)-positive), Tamoxifen (4-OHT) or Adeno-Cre induced recombination in AKP organoids after passage one (tdTomato and GFP positive) or sorted AKPE organoids (tdTomato positive), two passages after recombination (left). Depiction of genetic dual-reporter cassette (right). (C) PCR verification of genomic DNA from AKP plus vehicle, AKP plus Tamoxifen (4-OHT) or loading control (ctrl) samples for wild-type *Egfr*[+/+], floxed *Egfr*[fl/+] or recombined *Egfr*[Δ] alleles. (D) RT-qPCR analysis of *Egfr* mRNA in AKP, AKPE or AP organoids; $n = 4$, One-way ANOVA. (E) Absence of EGFR protein in the three independently derived EGFR-deleted AKPE organoids used in this study. (F) Quantification (left) of absolute GLUT1 protein amount between AKP, AKPE, and AP organoids assessed by western blot analysis (right), One-way ANOVA ($n = 4$, 3 or 4, respectively). (G) RT-qPCR analysis of *Slc2a1/Glut1* mRNA expression in AKP or AKPE organoids ($n = 7$). All data represent mean $+/-$ SEM. *P*-values calculated by paired, two-tailed t-test (between pairs of AKP and AKPE organoids) or One-way ANOVA.

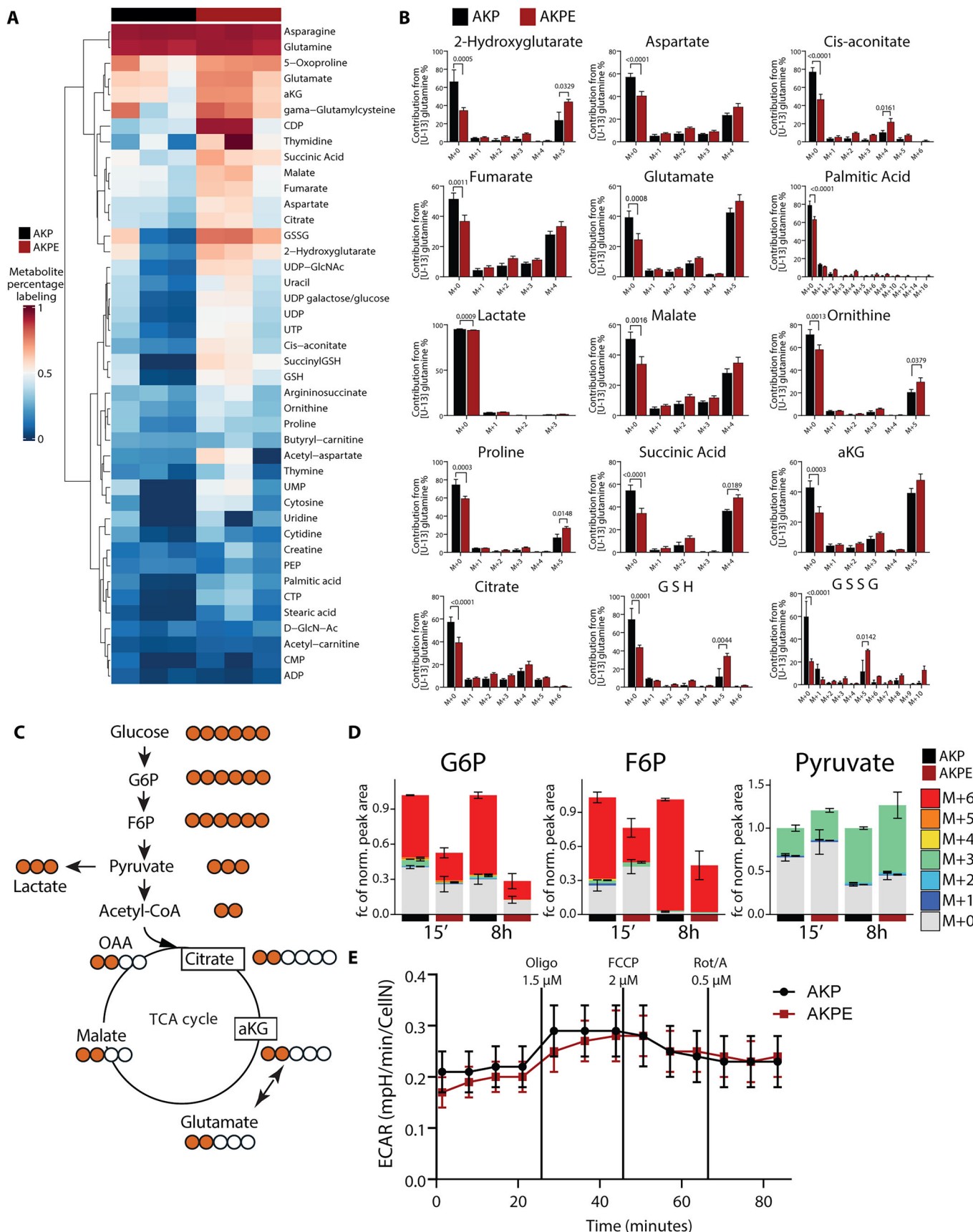

◀   **Figure EV2.   Stable isotope tracing and metabolic analysis reveals differential glutamine and glucose metabolism in EGFR-deficient organoids.**

(A) Heatmap showing fractional labeling of respective metabolites in AKP or AKPE organoids. The color scale corresponds to the z-score value of relative abundance of the metabolite. (B) Fractional enrichment in $^{13}$C-glutamine derived isotopologues of indicated metabolites as determined by LC-MS analysis. M + 0 (all carbons unlabeled) to M + n isotopologues indicate number of $^{13}$C atoms present in respective metabolite ($n = 3$ biological organoids with 4 technical replicates per organoid line).
(C) Schematic overview of glycolytic and oxidative catabolism and fractional contribution of labeled $^{13}$C carbons of indicated metabolites. (D) Glucose tracer metabolomics in AKPE ($n = 2$, biological) or AKP ($n = 2$, biological) organoids (technical replicates 6 per organoid line) showing fold-change at timepoints 15 min and 8 h of fractional labeling of glucose-6-phosphate (G6P), fructose-6-phosphate (F6P), pyruvate. M + 0 (all carbons unlabeled) to M + n isotopologues indicate number of $^{13}$C atoms present in respective metabolite. Total abundance normalized to protein (BCA) content. (E) Representative extracellular acidification rate (ECAR) measurement of AKP and AKPE organoids ($n = 3$, biological) obtained by the Mito Stress test of the Seahorse XF analysis. Oligomycin, FCCP, Rotenone (Rot) and antimycin-A (A) were added at indicated timepoints. aKG: alpha-ketoglutarate. G6P: Glucose-6-phosphate. TCA: tricarboxylic acid. All data represent mean $+/-$ SEM. *P*-values calculated by paired, two-tailed t-test (between pairs of AKP and AKPE organoids).

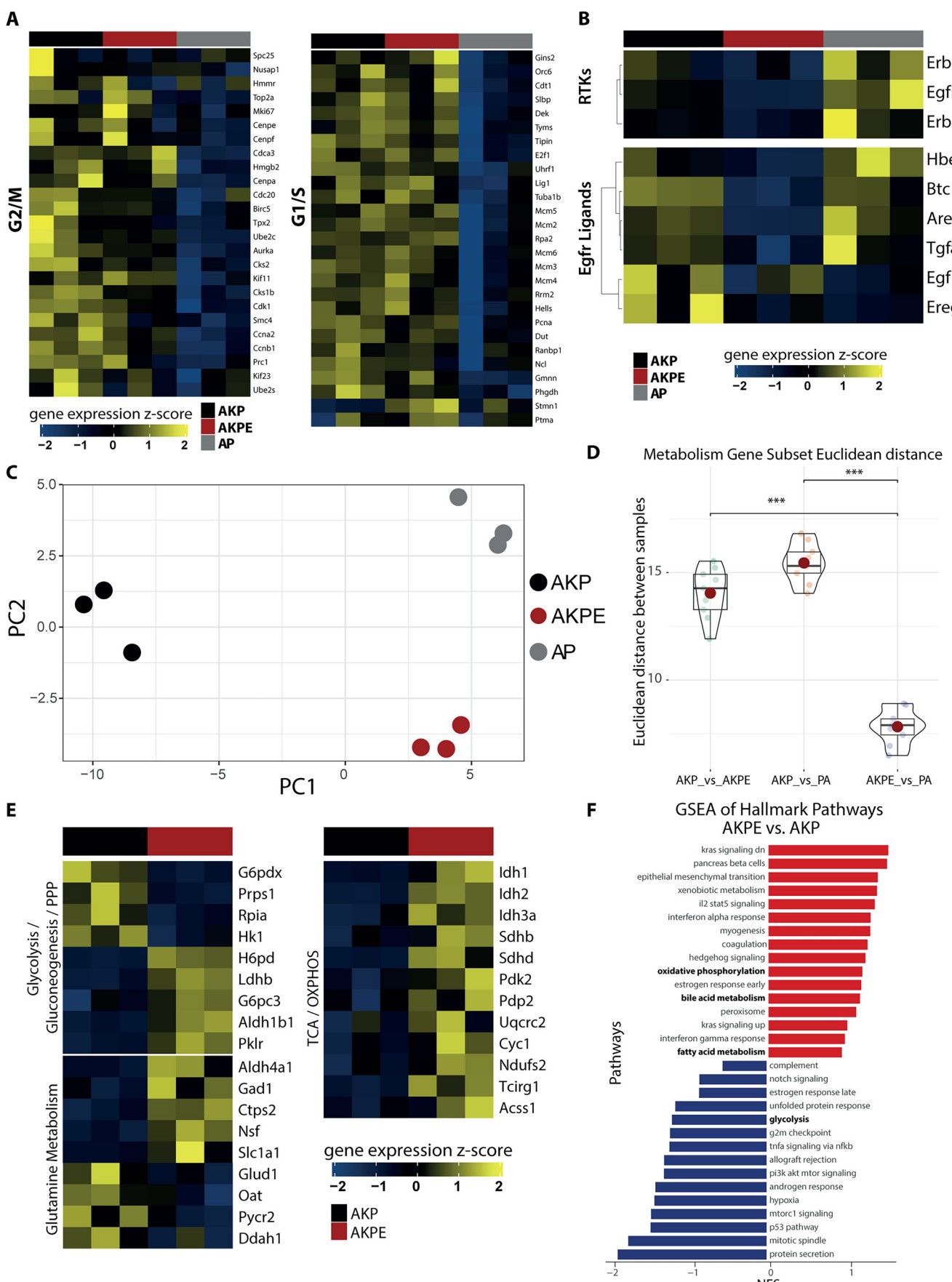

◀ **Figure EV3. Distinct transcriptional changes induced by KRAS^G12D expression and EGFR deletion.**

(A) Heatmap showing expression of selected cell cycle progression genes (based on (Avraham-Davidi et al, 2024)) of AKP, AKPE, and AP transcripts. (B) Heatmap showing expression of selected gene transcripts of receptor tyrosine kinases (RTKs) and EGFR ligands in AKPE, AKPE or AP organoids. (C) Principal component analysis (PCA) of metabolic gene subset of AKP, AKPE and AP organoids in steady state. Each subpopulation is depicted by indicated color. (D) Euclidean distance between samples of first and second principal component of metabolic gene subset between annotated comparisons, depicting comparisons of each sample to all other samples ($n = 9$). Horizontal lines denote the median and dots the mean. The box limits indicate 25th and 75th percentiles, whiskers extend to 1.5× of the interquartile range (IQR) from the 25th and 75th percentiles. (E) Heatmap showing expression of metabolic genes between AKP and AKPE organoids categorized according to molecular metabolic functional pathways. (F) Gene set enrichment analysis (GSEA) showing MSigDB Hallmark pathways significantly enriched in AKPE versus AKP ranked by their normalized enrichment score (NES). All data represent mean $+/-$ SEM. P-values calculated by One-way ANOVA.

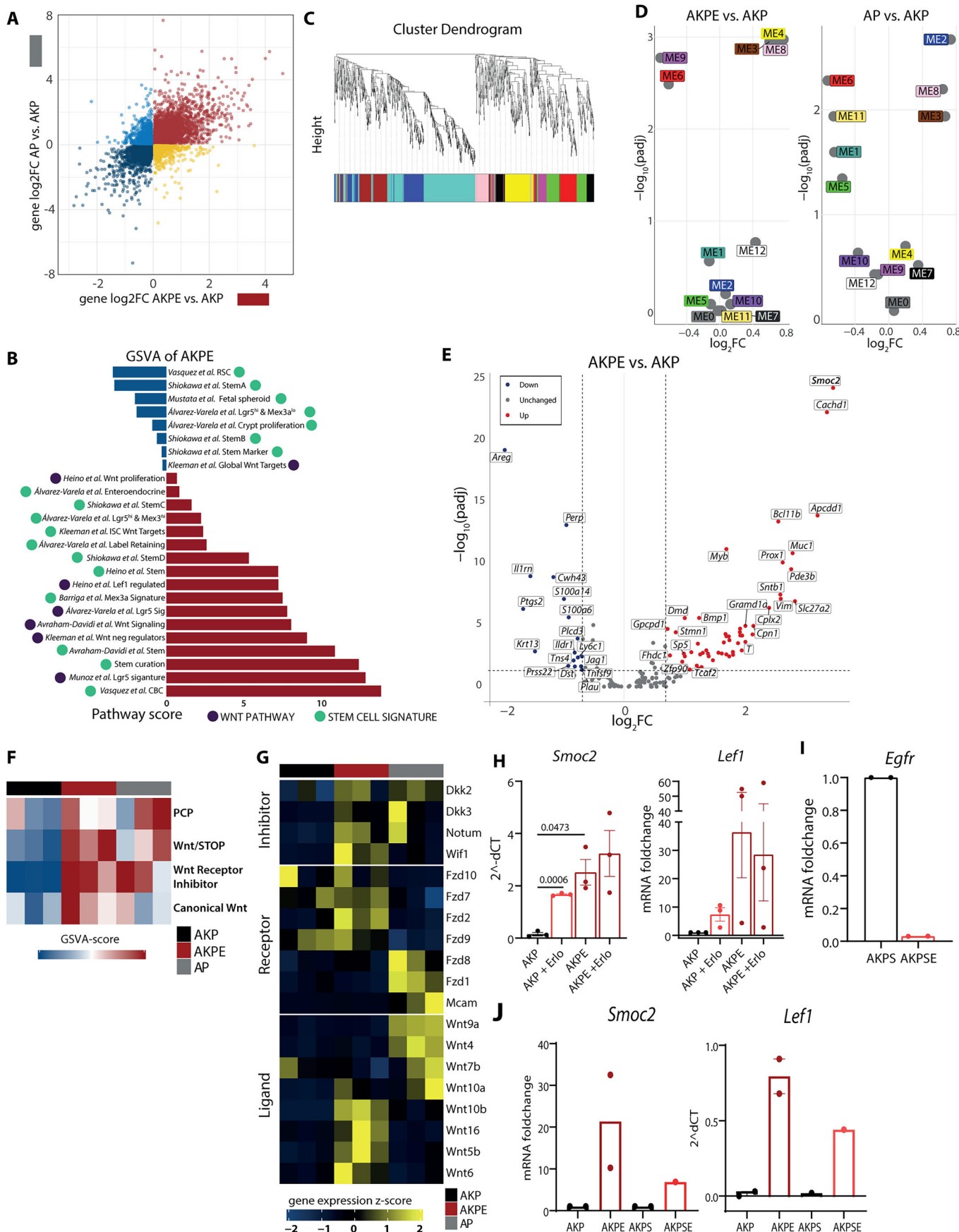

**Figure EV4.   Transcriptomic analysis reveals WNT pathway genes affected in EGFR-deficient colorectal organoids.**

(A) Scatter plot of fold changes from differential expression of AKPE versus AKP plotted against AP versus AKP derived fold changes. (B) Pathway enrichment of gene set variation analysis (GSVA) in AKPE organoids ranked by their pathway score. (C) Dendrogram of marker genes obtained by weighted gene correlation network analysis (WGCNA) according to modules. (D) Volcano plot of pathway-level modules derived from WGCNA analysis. (E) Volcano plot representing up or down-regulated gene expression of AKPE-associated genes according to fold-change ($n = 3$). (F) Heatmap of GSVA analysis of indicated WNT-pathways. (G) Heatmap of selected WNT-receptor and ligand interaction genes. (H) RT-qPCR analysis of *Smoc2* and *Lef1* mRNA expression in AKP, AKPE and erlotinib treated organoids ($n = 3$). (I, J) RT-qPCR analysis of *Egfr*, *Smoc2*, *Lef1* mRNA expression in AKPS or AKPSE organoids ($n = 2$). All data show mean $+/-$ SEM, or for I data mean $+/-$ SD. *P*-values calculated by paired, two-tailed t-test (between pairs of AKP and AKPE organoids).

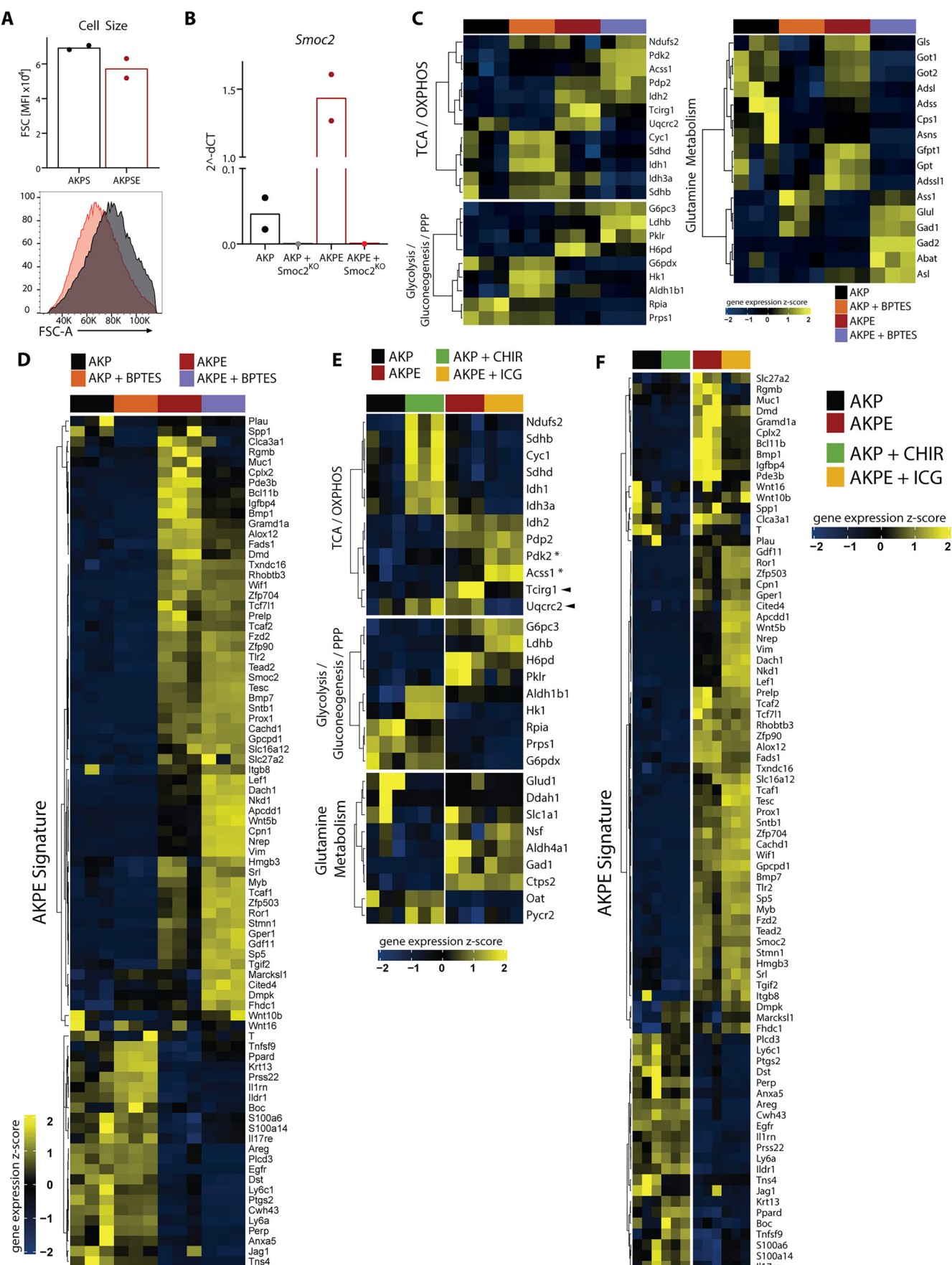

◀ **Figure EV5. Characterisation of organoids lacking Smoc2.**

(A) Forward scatter mean fluorescent intensity (MFI) and representative histogram of AKPS and AKPSE organoids assessed by flow cytometry. Data show mean $+/-$ SEM. (B) RT-qPCR analysis of *Smoc2* mRNA in AKP, AKP + Smoc2[KO] or AKPE and AKPE + Smoc2[KO] organoids ($n = 2$). (C) Heatmap showing subset of metabolic genes in AKP, AKPE or AKP and AKPE treated with control vehicle or 10 μM BPTES. (D) Heatmap showing expression of AKPE-signature genes in AKP, AKPE or AKP and AKPE treated with control vehicle or 10 μM BPTES. (E) Heatmap showing a subset of metabolic genes in AKP, AKPE, or AKP and AKPE treated with control vehicle or 3 μM CHIR99021 or 2 μM ICG-001. (F) Heatmap showing expression of AKPE-signature genes in AKP, AKPE, or AKP and AKPE treated with control vehicle or 3 μM CHIR99021 or 2 μM ICG-001.

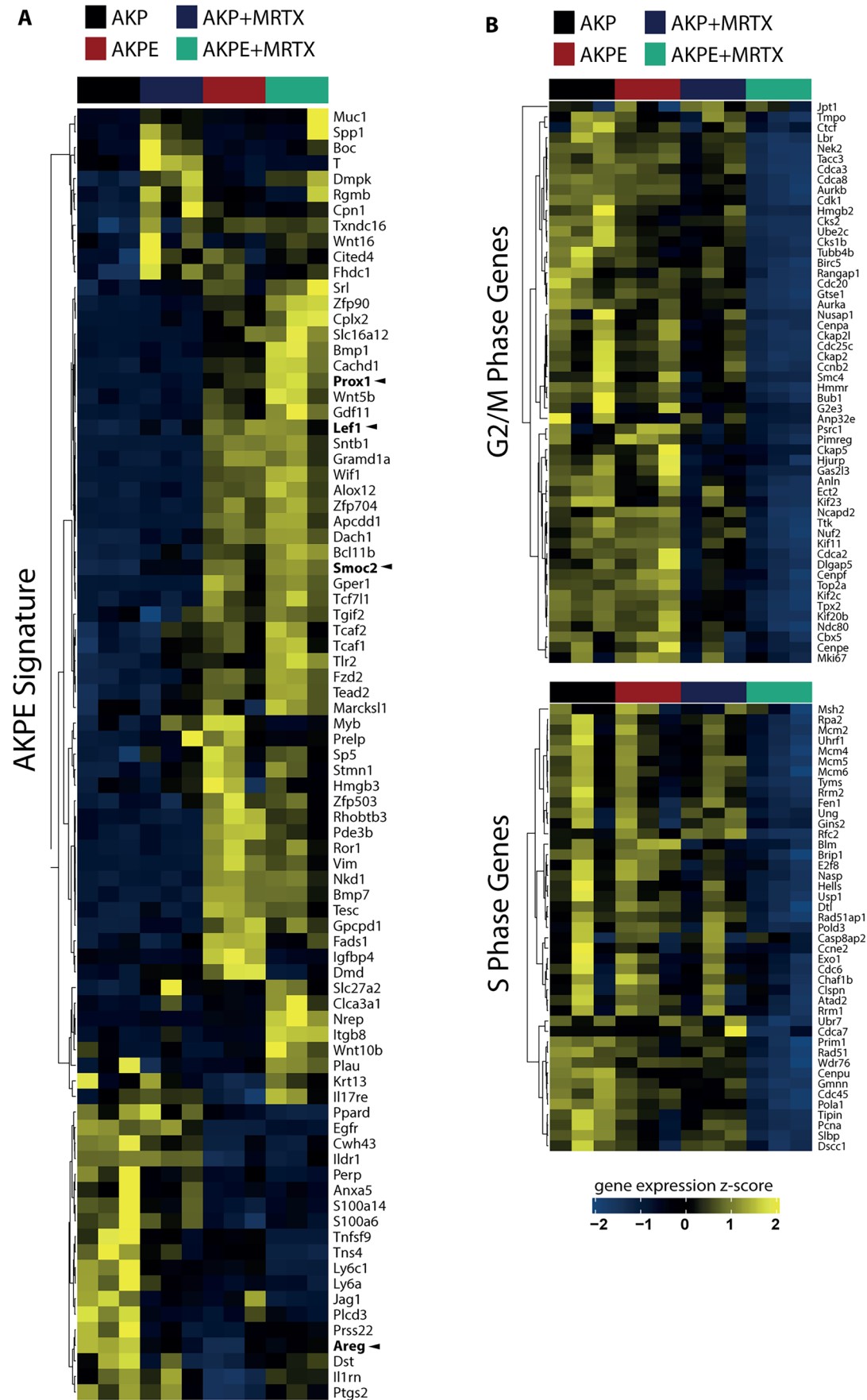

◀ **Figure EV6.    Distinct response to KRAS inhibition highlights EGFR loss as primary driver of AKPE gene signature.**

(**A**) Heatmap showing expression of AKPE-signature genes in AKP, AKPE, or AKP and AKPE treated with control vehicle or MRTX1133 (100 nM). (**B**) Heatmap showing expression of selected cell cycle progression genes in AKP, AKPE, or AKP and AKPE treated with control vehicle or MRTX1133 (100 nM).

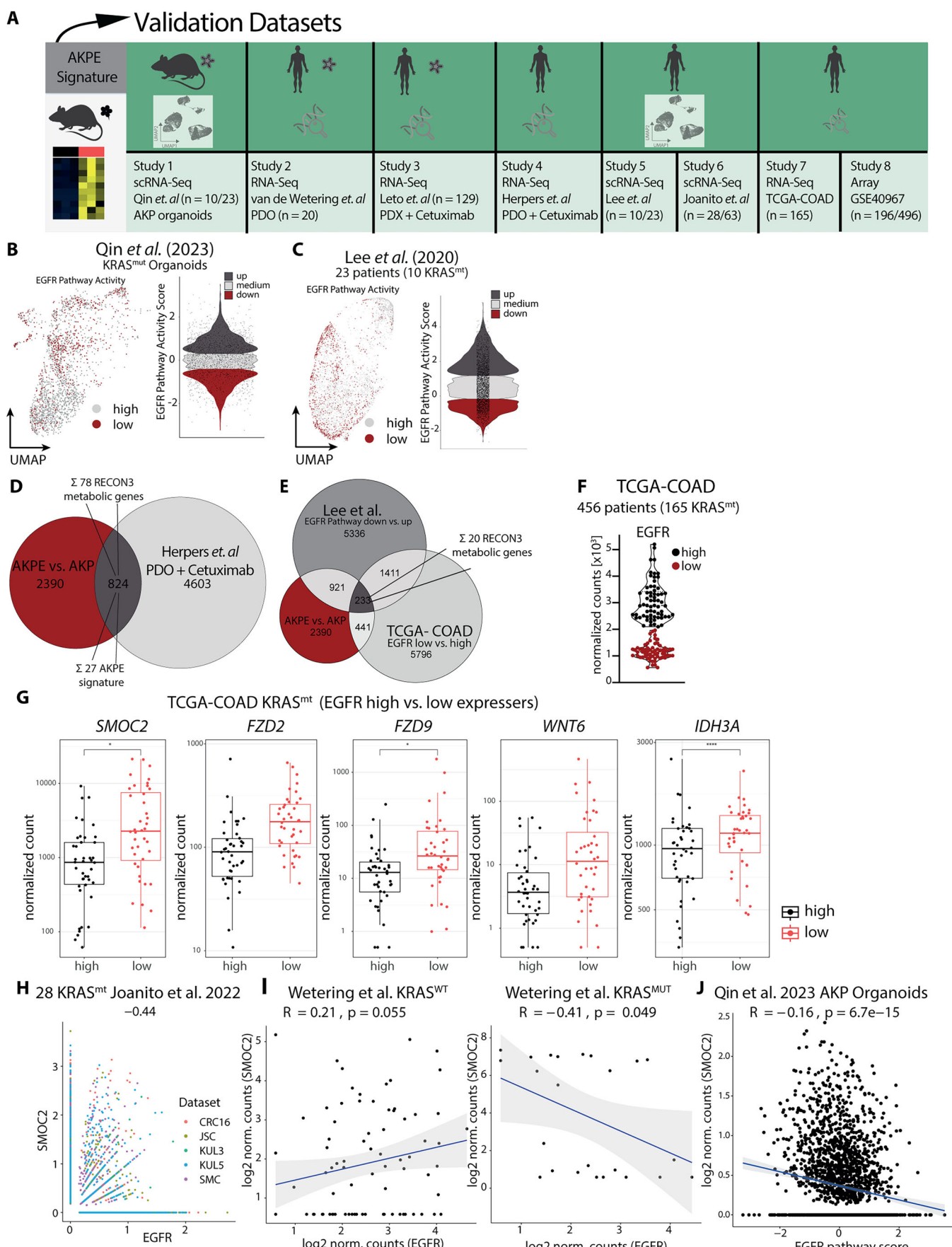

**Figure EV7.   Validation of EGFR–associated AKPE signatures across CRC datasets from mouse organoids and human tumors.**

(A) Graphical overview of accessed, publicly available mouse and human CRC datasets. *n* indicate number of KRAS^mt of all samples. (B) Classification of single cells based on EGFR high or low pathways activity from Qin et al, 2023 ($n = 10$ organoids). (C) Classification of single cells based on EGFR high or low pathways activity from Lee et al, 2020 ($n = 10$ patients). (D) Venn diagram of mutually expressed genes between AKPE versus AKP, EGFR high versus low expressors from Herpers et al PDOs treated with cetuximab or control vehicle. (E) Stratification of TCGA-COAD KRAS^mt patients into high and low *EGFR* expressors ($n = 165$ patients). (F) Venn diagram of mutually expressed genes between AKPE versus AKP, *EGFR* high vs. low expressors from TCGA-COAD and cells with down versus upregulated EGFR pathway signature from Lee et al, 2020. (G) Normalized counts of *SMOC2, FZD2, FZD9, WNT6* and *IDH3A* of TCGA-COAD KRAS^mt patients of *EGFR* high and low expressors ($n = 165$ patients). Horizontal lines denote the median, box limits indicate 25th and 75th percentiles and whiskers extend from the hinge to the lowest/largest value no further than 1.5x IQR from the 25th and 75th percentiles. (H) Scatter correlation plots of *EGFR* versus *SMOC2* expression in KRAS^mt single cells of the Joanito et al dataset. (I) Correlation of *EGFR* versus *SMOC2* expression in KRAS^wt or mt patients of the Wetering et al dataset. (J) Correlation of *EGFR* versus *SMOC2* in single cells of AKP organoids in the Qin et al dataset.

