## [Peer Review File · EMBO Molecular Medicine]

EGFR controls transcriptional and metabolic rewiring in KRASG12D colorectal cancer

Dana Krauss, Veronica Moreno-Viedma, Emi Adachi-Fernandez, Cristiano De Sa Fernandes, Jakob-Wendelin Genger, Ourania Fari, Bernadette Blauensteiner, Dominik Kirchhofer, Nikolina Bradaric, Valeriya Gushchina, Georgios Fotakis, Thomas Mohr, Ifat Abramovich, Inbal Mor, Martin Holcman, Andreas Bergthaler, Arvand Haschemi, Zlatko Trajanoski, Juliane Winkler, Eyal Gottlieb, and Maria Sibilja

Corresponding author: Maria Sibilja (maria.sibilja@meduniwien.ac.at)

Review Timeline:

Submission Date:	16th May 24
Editorial Decision:	14th Jun 24
Revision Received:	14th Jan 25
Editorial Decision:	30th Jan 25
Revision Received:	9th Mar 25
Editorial Decision:	13th Mar 25
Revision Received:	26th Mar 25
Accepted:	2nd Apr 25

Editor: Lise Roth

Transaction Report:

14th Jun 2024

Dear Maria,

Thank you for the submission of your manuscript to EMBO Molecular Medicine. We have now received feedback from the three reviewers who agreed to evaluate your manuscript.

As you will see from the enclosed reports, the referees acknowledge the potential interest of the findings and the interest of the organoid models, however they also raise major concerns, mostly related to the lack of demonstrated causality or experimental evidence that would provide the translational significance required for publication in EMM.

Based on the nature of the concerns and considering that at EMBO Press we encourage a single round of revisions in a limited time frame, we prefer to return the manuscript to you at this point with the decision that we cannot offer to publish it.

Given the potential interest of the findings, we would, however, be willing to consider a new manuscript on the same topic if at some time in the near future you obtained data that would considerably strengthen the message of the study and address the referees concerns in full. To be completely clear, however, I would like to stress that if you were to send a new manuscript this would be treated as a new submission rather than a revision and would be reviewed afresh, in particular with respect to the literature and the novelty of your findings at the time of resubmission. If you decide to follow this route, please make sure you nevertheless upload a letter of response to the referees' comments.

I am sorry that I could not bring better news this time and hope that the referee comments are helpful in your continued work in this area.

With kind regards,

Lise

**** Reviewer's comments ****

Referee #1 (Remarks for Author):

Krauss et al study the relevance of EGFR signaling in colon cancer cells carrying a mutant KRAS oncogene. It has long been known that EGFR inhibition in KRAS mutant CRC cells is ineffective, but that inhibition of KRAS and EGFR does lead to responses in patients. The authors establish a nice set of isogenic organoids to study the impact of EGFR deletion in colon cells that have lost APC and gained KRASG12D mutation. The authors show that loss of EGFR in this genetic background leads to metabolic rewiring from glycolysis to glutaminolysis. This is associated with gain of a stem cell signature and WNT activation. SMOC2 is identified as a critical target in this process.

Critique:

The data as presented reveal a number of associations without digging into the question of causality. It is suggested that Smoc2 is relevant to the induction of the WNT and CSC signatures in EGFR knockout cells, but this is not tested experimentally by suppressing Smoc2 using shRNA, CRISPi etc. Similarly, it is suggested that the metabolic phenotype occurring after EGFR deletion is driven by WNT signaling, but this is not tested. Would activation of WNT signaling by for example GSK3-beta inhibitors in the APK organoids indeed give the metabolic phenotype as observed with the EGFR deletion.

Or would WNT inhibitors prevent this metabolic phenotype upon EGFR deletion?

More relevant to the clinical responses to selective KRAS inhibitors is whether the WNT signaling and metabolic reprogramming are required for responses to these RAS inhibitors.

Establishing causal relationships and evaluating the contribution of these observed phenotypes to responses to KRAS inhibitors would be required to make this manuscript of interest to the readership of EMM.

Minor:

Figure 5B, the scale on the y-axis doesn't start at zero and the difference in fold change appears to be very minimal.

Referee #2 (Comments on Novelty/Model System for Author):

The colon cancer organoid model developed from GEMM mice with induced deletion of EGFR with KRAS G12D, p53 and APC mutations, is interesting.

The model could be improved by single cell transcriptional analysis that could describe how cell population are modulated by EGFR signalling.

Referee #2 (Remarks for Author):

In this MS, Kraus et al. shed light on the role of EGFR activation in colorectal tumors carrying mutations activating KRAS. They develop an organoid model from GEMM mice with induced deletion of EGFR in mice with KRAS G12D, p53 and APC mutations. The rationale as well the aim of the study is really well explained. Likewise, the presented results are original and moderately interesting.

Although AKPE vs AKP organoids do not show significant difference in growth and replication, the authors prove that in absence of EGFR signalling their metabolism partially shifted to more OXPHOS metabolism and to transcriptional program characterized by WNT gene cluster.

Albeit the MS have a high scientific quality, in my opinion it suffers of limitations due on side to lack of novelty and on the other to bulk transcriptional analysis.

Several reports suggest that EGFR signalling is related to WNT transcriptional program in colon cells. Moreover, the unbalance of these factors alters the amount of different cell populations present in gut and in organoids. These evidences support the hypothesis that phenotypic heterogeneity in cancer could lead drug resistance, including EGFR inhibitions. Therefore, a single cell analysis of the organoid models proposed would be much more informative.

The authors explore also the metabolic alterations in organoids, which indicate a reduction of glucose cell intake and some shift in energy metabolism. However, these results are not clearly and logically integrated with the other results and in the discussion. In others words, they appear as almost extraneous.

Moreover, no experiments of perturbation in altered metabolic pathways (e.g inhibition of glucose uptake and glutaminolysis) has been performed. Similarly, no experiments showing if metabolism alterations are causative of WNT populations are reported.

I suggest to improve this part and to motivate in a better way the role of metabolism changes both in Results and Discussion section.

Minor issues

-Fig.1B. Figure shows "doubling rate", however is not reported how is calculated. In Methods they mention the Incucyte analysis without specific details. However, Incucyte measures the projected area of organoids and possibly the number of organoids, so how is derived the doubling rate? Moreover, if the authors consider that cell size is reduced, how this feature can alters measurement with Incucyte?

-Fig.1C Replication rate is calculated with Ki67 by FACS analysis but the analysis of an indirect marker of cell replication that moreover is intracellular (not a good conditions for FACS analysis), instead of a EDU staining which effectively indicates the duplicate cells is quite singular.

-Fig 1E Graph reporting WB blot data is not proper. Normalizing the relative expression in this manner appears quite singular. The FC over housekeeping, reported in supplemental materials, is more correct. Moreover, the total level of Glut1 by WB is not much informative while the amount of Glut1 on plasma membrane is more indicative of increased intake of glucose. I suggest to evaluate the membrane expression of GLUT by FACS.

-both in abstract and discussion the authors say that glutaminolysis is increased but they show that glutamine uptake is not altered. So how they do explain this apparent paradox ?

- in addition they perform the LC-MS analysis using labelled glutamine without explaining the reason, particularly because they refer to mitochondrial metabolism changes. It should be explained better or possibly eliminated.

-in Results (line 222-224) authors say that others EGFR ligands are downregulated and this suggests the involvement of a paracrine loop with tumor microenvironment. Frankly speaking I do not understand why they say this. Please explain better this concept.

-In Discussion (lines 434-436), authors cited a review on WNT and metabolism. This review, together with their cited articles, show that cell with activated WNT program increases glycolysis and reduce OXPHOS metabolism, then it is not "consistently" with what it has been shown in the MS. Please discuss this issue.

Referee #3 (Comments on Novelty/Model System for Author):

The authors could use the information obtained to provide experimental evidence (both in vitro and in vivo) of the clinical relevance of their findings

Referee #3 (Remarks for Author):

The paper by Krau et al. focuses on the potential role of EGFR signaling in colorectal tumor cells harboring mutant KRAS. Although it is well known that EGFR inhibition is indicated only for patients carrying wild-type KRAS, the authors mention that there are few publications suggesting that EGFR inhibition might have some effect even in the presence of mutant KRAS, and they want to provide information on the putative mechanisms involved.

The work is based on the generation and analysis of three different organoid lines carrying either defective functions for APC/p53 (AP); APC/p53/KRAS MUT (AKP) or APC/p53/KRAS MUT+EGFR deletion (AKPE).

The authors first characterize these organoid lines in detail at the metabolic level and find a decreased use of glucose but a significant increase in the mitochondrial TCA pathway, which is relevant in the context of cancer. However, these results are only mentioned and not used as a basis for the following sections of the manuscript or at least there is no functional demonstration of its relevance in the context of EGFR inhibition/depletion. As an additional experimental test, there is an entire figure showing that KRAS mutant cells with EGFR deletion are smaller in size, but again, the relevance of this observation is not further explored. In fact, the entire body of the manuscript is based on a bulk RNA-seq analysis of AP, AKP and AKPE organoids (3 biological replicates each). From the RNA-seq data, the authors perform several bioinformatic analyses that ultimately uncover a transcriptomic signature that is specifically induced in AKPE. This signature includes elements of the WNT pathway and elements associated with intestinal stemness. As a positive comment, I have to say that the bioinformatic analysis of the data is well done, although there is nothing particularly innovative that alone would merit publication in EMM. Then the authors look at public datasets of colorectal cancer to demonstrate (with varying degrees of significance) that human tumors with reduced EGFR-related signaling have an increased AKPE signature. Based on the results provided, authors conclude that the AKPE signature and additional data obtained in this work could be useful in the future for the establishment of novel targeted therapies.

In general, I find that this work lacks experimental evidence to support the clinical relevance of the results obtained. This is not only related to the elements associated with the AKPE signature such as WNT or stemness-related genes such as Smoc2 (which is further validated by the authors as a gene/protein upregulated upon EGFR deletion, but not functionally), but also to the metabolic change towards mitochondrial respiration or the changes related to cellular compaction (decrease in cell size). Therefore, in the absence of clear experimental evidence that the result presented here could be applied to CRC patients, I do not think that this manuscript is a candidate for publication in EMBO Molecular Medicine.

As a service to authors, EMBO provides authors with the possibility to transfer a manuscript that one journal cannot offer to publish to another EMBO publication. The full manuscript and if applicable, reviewers reports are automatically sent to the receiving journal to allow for fast handling and a prompt decision on your manuscript. For more details of this service, and to transfer your manuscript to another EMBO title please click on Link Not Available

Referee #1 (Remarks for Author):

Krauss et al study the relevance of EGFR signaling in colon cancer cells carrying a mutant KRAS oncogene. It has long been known that EGFR inhibition in KRAS mutant CRC cells is ineffective, but that inhibition of KRAS and EGFR does lead to responses in patients. The authors establish a nice set of isogenic organoids to study the impact of EGFR deletion in colon cells that have lost APC and gained KRASG12D mutation. The authors show that loss of EGFR in this genetic background leads to metabolic rewiring from glycolysis to glutaminolysis. This is associated with gain of a stem cell signature and WNT activation. SMOC2 is identified as a critical target in this process.

We thank the reviewer for the constructive feedback, we have addressed his/her points as follows:

Critique:

The data as presented reveal a number of associations without digging into the question of causality. It is suggested that *Smoc2* is relevant to the induction of the WNT and CSC signatures in EGFR knockout cells, but this is not tested experimentally by suppressing *Smoc2* using shRNA, CRISPi etc.

This is indeed a highly relevant point. While our data suggest an association between SMOC2 and the induction of the WNT and CSC signatures in EGFR knockout cells, we have not experimentally demonstrated causality by directly suppressing *Smoc2* expression. We therefore utilized CRISPR-Cas9 technology to knockout SMOC2 in AKP and AKPE organoids. These new results are described in the new chapter on pages 13-15, lines 339-360 and 389-391. Fig. 5D-H and EV5B now represent data collected from these experiments, some of which are also shown below.

We demonstrate that knockout of *Smoc2* reverts the phenotype induced by EGFR deletion. RNA sequencing reveals that *Smoc2* knockout disrupts the induction of WNT and stemness-associated genes in EGFR-deficient organoids (AKPE), reverting the transcriptomic profile closer to EGFR wild-type organoids (AKP) (Fig. 5E, F). Additionally, metabolic gene analysis shows that *Smoc2* knockout selectively altered a subset of genes in EGFR-deleted organoids (Fig. 5H). The glucose uptake reduction observed in EGFR-deficient organoids is restored to wild-type levels when *Smoc2* is additionally deleted in AKPE organoids, while *Smoc2* knockout has no effect in AKP organoids (Fig. 5G). Moreover, the reduction in cell size is also rescued when SMOC2 is additionally deleted in AKPE organoids (Fig. 5D).

These findings strongly establish causality between EGFR deletion and the observed phenotypes identifying *SMOC2* as a key mediator in driving the observed metabolic changes, the induction of WNT and CSC transcriptional signatures as well as changes in cell size.

Similarly, it is suggested that the metabolic phenotype occurring after EGFR deletion is driven by WNT signaling, but this is not tested. Would activation of WNT signaling by for example GSK3-beta inhibitors in the AKP organoids indeed give the metabolic phenotype as observed with the EGFR deletion. Or would WNT inhibitors prevent this metabolic phenotype upon EGFR deletion?

We thank the reviewer for this important suggestion. To investigate the relationship between WNT signaling and the metabolic phenotype observed after EGFR deletion, we conducted additional experiments using a WNT activator (CHIR99021, a GSK3 β inhibitor) and a WNT pathway suppressor (ICG-001, a β -Catenin/TCF inhibitor) on the respective AKP and AKPE organoids followed by RNAseq analysis.

RNAseq analyses revealed that, while these treatments altered a small subset of metabolic genes in EGFR wild-type organoids, they could neither replicate nor prevent the unique metabolic gene expression signature in EGFR-deleted organoids (Fig. EV5E). Furthermore, in EGFR-deleted organoids, only a small subset of genes within the AKPE signature showed suppression compared to wildtype AKP organoids (Fig. EV5F). Combined with findings from *Smoc2* knockout organoids, these results indicate that the metabolic phenotype induced by EGFR deletion is not solely driven by WNT signaling but involves broader regulatory networks that depend on EGFR signaling. EGFR deletion likely disrupts these networks, leading to compensatory changes or direct effects on metabolic regulation.

In conclusion, while WNT signaling plays a role in metabolic regulation, in our experimental system with KRAS mutation, the metabolic phenotype associated with EGFR deletion appears to result from a combination of altered WNT signaling and additional EGFR-dependent pathways. This phenotype cannot be fully rescued by either WNT pathway activators or inhibitors. We appreciate the reviewer's suggestion to explore this further, as it has clarified that the observed metabolic alterations in our system are not primarily driven by WNT pathway changes but rather by SMOC2 which is upregulated in the absence of EGFR. We have updated our manuscript to reflect this understanding on pages 14-15, lines 372-388, page 22, lines 571-576 and in Fig. EV5E,F.

More relevant to the clinical responses to selective KRAS inhibitors is whether the WNT signaling are required for responses to these RAS inhibitors. Establishing causal relationships and evaluating the contribution of these observed phenotypes to responses to KRAS inhibitors would be required to make this manuscript of interest to the readership of EMM.

We thank the reviewer for this important suggestion. To address this point, we treated AKP and AKPE organoids with the selective KRAS^{G12D} inhibitor MRTX1133. These results are described on page 17-18, lines 432-462, page 22, lines 583-586, 591-595 and shown in Fig. 7 and Fig. EV7.

While MRTX1133 had minimal effects on proliferation in EGFR wild-type (AKP) organoids, it completely inhibited proliferation in EGFR-deficient (AKPE) organoids (Fig. 7A). RNA sequencing revealed that KRAS inhibition modestly increased the AKPE signature score in AKP organoids but did not activate the WNT pathway (Fig. 7B, EB7A). WNT and stemness signatures, along with key target genes (e.g., Smoc2, Areg, Prox1, Lef1), were specifically and uniquely upregulated in EGFR knockout organoids (Fig. 7B, EB7A). PROGENy analysis confirmed that EGFR loss uniquely drives WNT response genes, unaffected by KRAS inhibition (Fig. 7A). Additionally, while EGFR deletion alone did not alter cell cycle genes, MRTX1133 reduced their expression in both AKP and AKPE organoids (Fig. 7D, EV7B).

These findings underscore the distinct roles of KRAS and EGFR in regulating tumorigenesis and significantly enhance the clinical relevance of our study. Firstly, the essentially of dual inhibition of EGFR and KRAS, as KRAS inhibition alone only modestly reduces proliferation, whereas dual inhibition completely inhibits it. But more importantly, we can show that the induction of the WNT- and stemness signature is uniquely driven by EGFR loss and cannot be replicated by KRAS inhibition, thus suggesting that it may be a prerequisite for effective response to KRAS inhibitors. As MRTX1133 is currently in Phase 1/2 clinical trials (NCT05737706), we think that our results align well with the interests of the EMM readership, highlighting the potential implications for therapeutic strategies.

Minor:

Figure 5B, the scale on the y-axis doesn't start at zero and the difference in fold change appears to be very minimal. causality.

We thank the reviewer for pointing this out. We have adjusted the y-axis scale to start at zero as suggested. The updated figure is now included as Figure 4G.

Referee #2 (Comments on Novelty/Model System for Author):

The colon cancer organoid model developed from GEMM mice with induced deletion of EGFR with KRAS G12D, p53 and APC mutations, is interesting. The model could be improved by single cell transcriptional analysis that could describe how cell population are modulated by EGFR signaling.

Referee #2 (Remarks for Author):

In this MS, Kraus et al. shed light on the role of EGFR activation in colorectal tumors carrying mutations activating KRAS. They develop an organoid model from GEMM mice with induced deletion of EGFR in mice with KRAS G12D, p53 and APC mutations. The rationale as well the aim of the study is really well explained. Likewise, the presented results are original and moderately interesting. Although AKPE vs AKP organoids do not show significant difference in growth and replication, the authors prove that in absence of EGFR signaling their metabolism partially shifted to more OXPHOS metabolism and to transcriptional program characterized by WNT gene cluster. Albeit the MS have a high scientific quality, in my opinion it suffers of limitations due on side to lack of novelty and on the other to bulk transcriptional analysis.

We thank the reviewer for the positive comments on our study. All the raised issues were addressed as follows:

Several reports suggest that EGFR signaling is related to WNT transcriptional program in colon cells. Moreover, the unbalance of these factors alters the amount of different cell populations present in gut and in organoids. These evidences support the hypothesis that phenotypic heterogeneity in cancer could lead drug resistance, including EGFR inhibitions. Therefore, a single cell analysis of the organoid models proposed would be much more informative.

We agree that single-cell transcriptional analysis would provide valuable insights into how cell populations are modulated by EGFR signaling and improve the overall understanding of phenotypic heterogeneity in our organoid models. Therefore, we have followed the reviewers' suggestion and performed single-cell RNA sequencing on our organoid models. We added the resulting data in the manuscript in Fig. 6, EV6, pages 15-16 and lines 392-430, page 23, lines 605-615 and some data are shown below.

Our analysis reveals that EGFR deletion leads to a distinct shift in both cell population composition and differentiation status between AKP and AKPE organoids. Overall, we can confirm that EGFR deletion upregulates *Smoc2*, *Ctnnb1* (β -Catenin) (Fig. EV6A,B) and the specific AKPE signature (Fig. 6, EV6A,B). Simultaneously, canonical differentiation markers (e.g. *Krt20* or *Muc2*) are absent (Fig. 6D). Specifically, AKP organoids exhibit a more heterogeneous mixture of cell populations (Fig. 6C), characterized by the expression of distinct markers of stemness e.g. *Ascl2*, *Agr2*, *Lrig1* or *Lgr5* (Fig. 6A,B). In contrast, AKPE organoids

are notably more homogenous in their cellular composition, predominantly expressing stemness genes as *Prox1* or *Smoc2* (Fig. 6D). Cytotrace2 potency analysis highlights that AKPE organoids are characterized by higher potency scores and are enriched in multi- and oligopotent cells (Fig. 6F).

Overall, these observations substantiate the fundamental differences in the cellular dynamics and differentiation potential of AKP and AKPE organoids, offering insights into their developmental pathways and functional properties.

The authors explore also the metabolic alterations in organoids, which indicate a reduction of glucose cell intake and some shift in energy metabolism. However, these results are not clearly and logically integrated with the other results and in the discussion. In others words, they appear as almost extraneous. Moreover, no experiments of perturbation in altered metabolic pathways (e.g inhibition of glutaminolysis) has been performed. Similarly, no experiments showing if metabolism alterations are causative of WNT populations are reported. I suggest to improve this part and to motivate in a better way the role of metabolism changes both in Results and Discussion section.

We thank the reviewer this valuable comment. We have addressed his/her comments as follows:

We have improved the verbal integration of the already presented metabolic data (page 7, lines 164-166) and by performing additional experiments. We used inhibitors of glutaminolysis, BPTES, which specifically blocks the conversion of glutamine to glutamate and its downstream metabolites, and evaluated their effects on WNT signaling. RNAseq analysis revealed genotype-specific upregulation of metabolic genes in AKP and AKPE organoids (Fig. EV5C). However, BPTES treatment did not affect the WNT/stemness signature in EGFR-deleted

organoids (Fig. EV5D). These results are described on page 14 lines 361-371 and inserted below.

Additionally, as mentioned in our response to Reviewer 1, we treated AKP and AKPE organoids with WNT inhibitors and also deleted SMOC2 in AKP/AKPE organoids and assessed the corresponding changes in metabolic pathways (see response to reviewer 1 for more details). We show that SMOC2 is a key mediator in driving the observed metabolic changes, as glucose uptake upon SMOC2 deletion is rescued. However, the observed metabolic alterations induced by EGFR deletion do not seem to be primarily driven by WNT pathway changes, but involve additional EGFR-dependent pathways.

Altogether, these new results strengthen the link between EGFR deletion and the effect on energy metabolism and also link the observed metabolic changes to the targets identified.

Minor issues

- Fig.1B. Figure shows "doubling rate", however is not reported how is calculated. In Methods they mention the Incucyte analysis without specific details. However, Incucyte measures the projected area of organoids and possibly the number of organoids, so how is derived the doubling rate? Moreover, if the authors consider that cell size is reduced, how this feature can alters measurement with Incucyte?

We thank the reviewer for pointing out the need for further clarification on the calculation of the doubling rate. We have now added a more detailed explanation of how the doubling rate was calculated, which was based on an experiment independent from the Incucyte analysis. We have added this in the method section as follows (pages 24-25, lines 648-655):

“To assess the doubling rate, cells were passaged for a minimum of four consecutive passages. At each passage, cells were collected using 0.05% trypsin-EDTA for 20-40 minutes until single-cell suspensions were achieved. Cells were manually counted with 1:1 diluted trypan blue to assess cell viability with a hemocytometer. After counting, cells were seeded at a density of $10^4/100 \mu\text{L}$ Matrigel into one well of a 6-well plate. For assessment of doubling rate, final cell number over initial seeded cell number was calculated and averaged over the total passage number for each organoid line.”

We also acknowledge the reviewer's question regarding the impact of decreased cell size by the Incucyte measurements. While the Incucyte primarily measures the projected area of organoids and possibly their number, we observed that despite consistent proliferation rates (as assessed by Ki67, EdU, and proliferation genes by RNA sequencing), the organoids appeared smaller in size, which was reflected in the Incucyte measurements. These results suggested to us that the reduction in organoid area observed with Incucyte analysis is likely due to the decrease in individual cell size rather than a direct reduction in the number of cells or proliferation rate.

- Fig.1C Replication rate is calculated with Ki67 by FACS analysis but the analysis of an indirect marker of cell replication that moreover is intracellular (not a good conditions for FACS analysis), instead of a EDU staining which effectively indicates the duplicate cells is quite singular.

We agree that using EdU staining, which directly measures DNA synthesis, more accurately reflects cell replication. We have therefore performed the EdU staining experiment and the results confirm that there is no significant change in proliferation between AKPE and AKP organoids. These results are included in Figure 1C and mentioned in results page 6, lines 135-136.

- Fig 1E Graph reporting WB blot data is not proper. Normalizing the relative expression in this manner appears quite singular. The FC over housekeeping, reported in supplemental materials, is more correct. Moreover, the total level of Glut1 by WB is not much informative while the amount of Glut1 on plasma membrane is more indicative of increased intake of glucose. I suggest to evaluate the membrane expression of GLUT by FACS.

We thank the reviewer for the suggestions.

We have revised the representation of Figure 1E by changing the fold change (FC) from AKPE vs. AKP to relative GLUT1 expression, normalized over the housekeeping control. This should provide a more accurate representation of the data.

Additionally, we agree with the reviewer that assessing GLUT1 localization on the plasma membrane would provide more relevant information. We tried two different batches of the only available GLUT1 antibody targeting the extracellular domain (AGT-041, Alomone Labs, LTD). However, with this antibody we were unable to detect a specific signal by FACS analysis as can be seen in the figure below (left).

In order to answer the raised point, we performed GLUT1 immunofluorescence staining to compare membrane-bound versus intracellular GLUT1 expression. We segmented immunofluorescent stained cells into nucleus and membrane areas (based on co-stain with wheat germ agglutinin membrane and DAPI nucleus stain) and quantified the GLUT1 signal, respectively. We observe a decrease of GLUT1 signal in the membrane compartment of stained cells (see below, right). We therefore conclude that the overall protein amount, as measured by western blot is mainly driven by a decrease of GLUT1 protein on the plasma membrane. These results are shown in Fig. 1F and mentioned on page 7, lines 146-147.

- both in abstract and discussion the authors say that glutaminolysis is increased but they show that glutamine uptake is not altered. So how they do explain this apparent paradox?
- in addition they perform the LC-MS analysis using labelled glutamine without explaining the reason, particularly because they refer to mitochondrial metabolism changes. It should be explained better or possibly eliminated.

We thank the reviewer for pointing out this apparent paradox and for raising important questions about our interpretation of glutaminolysis and glutamine uptake. We acknowledge that the discrepancy between the unchanged glutamine uptake and the increased glutaminolysis could be puzzling. This paradox was indeed something we also found intriguing, and to our opinion can be explained by the differential utilization of glutamine inside the cells. While the overall glutamine uptake might remain constant, it is possible that glutamine is being used more efficiently or diverted into specific metabolic pathways, such as glutaminolysis, once inside the cell. This, can explain the observed increase in glutaminolysis despite no change in glutamine uptake.

We used LC-MS analysis with labeled glutamine to trace the fate of glutamine in the cells, especially in light of the reduced glucose uptake observed. Our primary goal was to understand the compensatory mechanisms that might be driving the metabolic shifts, particularly the increased reliance on glutamine metabolism. By tracing labeled glutamine, we aimed to gain insights into how cells might adapt to lower glucose availability by utilizing alternative carbon sources, particularly in mitochondrial metabolism.

We further clarified and explained the rationale for the use of labeled glutamine in our manuscript (page 7, lines 164-166). We hope that we have provided a more comprehensive explanation of these metabolic changes.

- in Results (line 222-224) authors say that others EGFR ligands are downregulated and this suggests the involvement of a paracrine loop with tumor microenvironment. Frankly speaking I do not understand why they say this. Please explain better this concept.

We agree that this section needs more explanation and have adjusted the passage accordingly. Here is the explanation for our formulation of paracrine effects: We previously discovered in GEMMs of CRC that EGFR deletion in myeloid cells reduces CRC development, while EGFR deletion in tumor cells has no effect on tumor growth (Srivatsa *et al*, 2017). This is a very surprising finding and we found EGFR also to be upregulated in myeloid cell infiltrating tumors of metastatic CRC patients suggesting that therapeutically in patients EGFR inhibitors might be effective by targeting myeloid cells. However, by targeting EGFR expressed on tumor cells, EGFR inhibitors might suppress the production of EGFR ligands, which would activate EGFR-positive myeloid cells present in the tumor microenvironment (TME). Therefore, EGFR might inhibit CRC growth by two mechanisms: first by directly inhibiting EGFR-positive myeloid cells, second by inhibiting the production of EGFR ligands in tumor cells and thus directly interfering with and autocrine proliferation loop and/or indirectly preventing the activation of EGFR-positive cells in the TME.

To avoid confusion, we have changed paracrine to autocrine in the result section (page 9, line 229), as there is no TME present in organoids and added an additional section to the discussion (page 23, lines 595-604).

- In Discussion (lines 434-436), authors cited a review on WNT and metabolism. This review, together with their cited articles, show that cell with activated WNT program increases glycolysis and reduce OXPHOS metabolism, then it is not "consistently" with what it has been shown in the MS. Please discuss this issue.

We thank the reviewer for raising this important point regarding a possible discrepancy between the cited review and our results.

In our study, we demonstrate that an altered metabolic state upon EGFR deletion in KRASG12D cells is accompanied by the upregulation of a distinct WNT and cancer stemness signature. This was particularly intriguing to us, as cancer stem cells (CSCs) are known to exhibit a distinct metabolism, which is distinct from differentiated cells. Our results highlight increased glutaminolysis in EGFR-deleted organoids, in parallel with an increase in stemness signatures, suggesting that these metabolic changes may play a role in maintaining or enhancing the cancer stem cell phenotype.

We acknowledge the point made in the cited review that WNT activation typically increases glycolysis and reduces oxidative phosphorylation (OXPHOS) in some contexts. However, we argue that the relationship between WNT signaling and metabolism is more nuanced, particularly when considering the varied metabolic needs of different stem cell populations. In healthy stem cells, processes like proliferation, quiescence, or cell death are influenced by metabolic states, which can vary depending on tissue type or even within distinct niches in the tissue (Beumer & Clevers, 2024; Jackson & Finley, 2024). For example, in the intestine, CBCs (crypt base columnar cells) rely on mitochondrial OXPHOS, while surrounding Paneth cells use increased glycolysis (Rodríguez-Colman et al., 2017). This indicates that different cells within the same tissue can exhibit compartmentalized metabolic programs, even when both are influenced by WNT signaling. We have revised the discussion to further clarify these points and emphasize the compartmentalized nature of metabolic pathways, particularly in the context of cancer stem cells and their unique metabolic demands, which may not always follow the canonical WNT-induced glycolysis model observed in other cell types page 22, lines 572-576.

Referee #3 (Comments on Novelty/Model System for Author):

The authors could use the information obtained to provide experimental evidence (both in vitro and in vivo) of the clinical relevance of their findings

Referee #3 (Remarks for Author):

The paper by Krauß et al. focuses on the potential role of EGFR signaling in colorectal tumor cells harboring mutant KRAS. Although it is well known that EGFR inhibition is indicated only for patients carrying wild-type KRAS, the authors mention that there are few publications suggesting that EGFR inhibition might have some effect even in the presence of mutant KRAS, and they want to provide information on the putative mechanisms involved.

The work is based on the generation and analysis of three different organoid lines carrying either defective functions for APC/p53 (AP); APC/p53/KRASMUT (AKP) or APC/p53/KRASMUT+EGFR deletion (AKPE). The authors first characterize these organoid lines in detail at the metabolic level and find a decreased use of glucose but a significant increase in the mitochondrial TCA pathway, which is relevant in the context of cancer. However, these results are only mentioned and not used as a basis for the following sections of the manuscript or at least there is no functional demonstration of its relevance in the context of EGFR inhibition/depletion.

We agree with the reviewer and apologize for not having integrated the metabolic analysis with the rest of the manuscript. This point was also raised by Rev. 2. We have performed more mechanistic experiments by metabolism perturbation, SMOC2 deletion and WNT inhibition in AKP and AKPE organoids followed by RNAseq analysis. With these new results the manuscript has been significantly improved mechanistically and the different parts are better linked to each other.

As an additional experimental test, there is an entire figure showing that KRAS mutant cells with EGFR deletion are smaller in size, but again, the relevance of this observation is not

further explored. In fact, the entire body of the manuscript is based on a bulk RNA-seq analysis of AP, AKP and AKPE organoids (3 biological replicates each). From the RNA-seq data, the authors perform several bioinformatic analyses that ultimately uncover a transcriptomic signature that is specifically induced in AKPE. This signature includes elements of the WNT pathway and elements associated with intestinal stemness.

Also in this context, we have performed more experiments as can be seen in the revised manuscript. We have also performed scRNAseq analysis and can confirm what we found by RNAseq, additionally allowing us to investigate cellular heterogeneity in the different organoids. See also responses to Rev. 1 and 2.

As a positive comment, I have to say that the bioinformatic analysis of the data is well done, although there is nothing particularly innovative that alone would merit publication in EMM. Then the authors look at public datasets of colorectal cancer to demonstrate (with varying degrees of significance) that human tumors with reduced EGFR-related signaling have an increased AKPE signature. Based on the results provided, authors conclude that the AKPE signature and additional data obtained in this work could be useful in the future for the establishment of novel targeted therapies.

We appreciate that the reviewer acknowledges the quality of our bioinformatic analysis, but disagree with the reviewer that there is nothing particularly innovative. To our knowledge, such an analysis has never been done before, looking specifically at Ras mutation in the context of EGFR deletion. Following the comments of the first 2 reviewers, we have performed a lot of mechanistic experiments, including scRNAseq, and hope that the reviewer will look more positively at our paper. See response to Rev. 1 & 2.

In general, I find that this work lacks experimental evidence to support the clinical relevance of the results obtained. This is not only related to the elements associated with the AKPE signature such as WNT or stemness-related genes such as Smoc2 (which is further validated by the authors as a gene/protein upregulated upon EGFR deletion, but not functionally), but also to the metabolic change towards mitochondrial respiration or the changes related to cellular compaction (decrease in cell size). Therefore, in the absence of clear experimental evidence that the result presented here could be applied to CRC patients, I do not think that this manuscript is a candidate for publication in EMBO Molecular Medicine.

As specified in our answers to Rev. 1 and 2, we have performed multiple functional experiments, including studies with Ras inhibitors which should address also these concerns of the reviewer.

30th Jan 2025

Dear Maria,

Thank you for submitting your revised study. We have now received the reports from the referees who had evaluated the original version of your manuscript. As you will see below, while referees #1 and #2 are satisfied with the revisions, referee #3 still raises a number of concerns on your manuscript.

We have discussed these concerns further with the referees, as well as within the team, and agreed that only few points should be addressed experimentally, whereas most could be addressed by additional discussion, toning down the claims and clarifying the flow of the manuscript.

We would therefore like to invite you to revise the manuscript further to address the remaining concerns.

In particular, we would like you to address experimentally the following points:

- Demonstration that only membrane-bound GLUT1 is reduced in AKPE cells, but not the nuclear fraction OR tone down this claim.
- Figure 5, AKPE organoids smaller than AKP organoids: Ki67 staining and/or cell cycle analysis.
- single cell RNA-seq of AKP and AKPE: sub-clustering in UMAP.
- Figure 7: formal demonstration of synergy OR tone-down.

We require:

- 1) A .docx formatted version of the manuscript text (including legends for main figures, EV figures and tables). Please make sure that the changes are highlighted to be clearly visible.
- 2) Individual production quality figure files as .eps, .tif, .jpg (one file per figure). For guidance, download the 'Figure Guide PDF' (<https://www.embopress.org/page/journal/17574684/authorguide#figureformat>).
- 3) At EMBO Press we ask authors to provide source data for the main figures. Our source data coordinator will contact you to discuss which figure panels we would need source data for and will also provide you with helpful tips on how to upload and organize the files.
- 4) A .docx formatted letter INCLUDING the reviewers' reports and your detailed point-by-point responses to their comments. As part of the EMBO Press transparent editorial process, the point-by-point response is part of the Review Process File (RPF), which will be published alongside your paper.
- 5) A complete author checklist, which you can download from our author guidelines (<https://www.embopress.org/page/journal/17574684/authorguide#submissionofrevisions>). Please insert information in the checklist that is also reflected in the manuscript. The completed author checklist will also be part of the RPF.
- 6) All Materials and Methods need to be described in the main text using our 'Structured Methods' format. According to this format, the Methods section includes a Reagents and Tools Table (listing key reagents, experimental models, software and relevant equipment and including their sources and relevant identifiers) followed by a Methods and Protocols section describing the methods, ideally using a step-by-step protocol format. The aim is to facilitate adoption of the methodologies across labs. Please download and fill our Reagents and Tools Table template (.docx), which you can find in our author guidelines: <https://www.embopress.org/page/journal/14693178/authorguide#structuredmethods>. When submitting your revised manuscript, please do not include the Reagents and Tools Table in the Methods section of the manuscript but upload it as a separate file choosing the file type "Reagent Table". An example of a Method paper with Structured Methods can be found here: <https://www.embopress.org/doi/10.15252/msb.20178071>
- 7) Please note that all corresponding authors are required to supply an ORCID ID for their name upon submission of a revised manuscript.
- 8) It is mandatory to include a 'Data Availability' section after the Materials and Methods. Before submitting your revision, primary datasets produced in this study need to be deposited in an appropriate public database, and the accession numbers and database listed under 'Data Availability'. Please remember to provide a reviewer password if the datasets are not yet public (see <https://www.embopress.org/page/journal/17574684/authorguide#dataavailability>).

In case you have no data that requires deposition in a public database, please state so in this section.

9) For data quantification: please specify the name of the statistical test used to generate error bars and P values, the number (n) of independent experiments (specify technical or biological replicates) underlying each data point and the test used to calculate p-values in each figure legend. The figure legends should contain a basic description of n, P and the test applied. Graphs must include a description of the bars and the error bars (s.d., s.e.m.). Please provide exact p values.

10) Our journal encourages inclusion of *data citations in the reference list* to directly cite datasets that were re-used and obtained from public databases. Data citations in the article text are distinct from normal bibliographical citations and should directly link to the database records from which the data can be accessed. In the main text, data citations are formatted as follows: "Data ref: Smith et al, 2001" or "Data ref: NCBI Sequence Read Archive PRJNA342805, 2017". In the Reference list, data citations must be labeled with "[DATASET]". A data reference must provide the database name, accession number/identifiers and a resolvable link to the landing page from which the data can be accessed at the end of the reference. Further instructions are available at .

11) We replaced Supplementary Information with Expanded View (EV) Figures and Tables that are collapsible/expandable online. A maximum of 5 EV Figures can be typeset. EV Figures should be cited as 'Figure EV1, Figure EV2' etc... in the text and their respective legends should be included in the main text after the legends of regular figures.

12) The paper explained: EMBO Molecular Medicine articles are accompanied by a summary of the articles to emphasize the major findings in the paper and their medical implications for the non-specialist reader. Please provide a draft summary of your article highlighting

13) Author contributions: CRedit has replaced the traditional author contributions section because it offers a systematic machine readable author contributions format that allows for more effective research assessment. Please remove the Authors Contributions from the manuscript and use the free text boxes beneath each contributing author's name in our system to add specific details on the author's contribution. More information is available in our guide to authors.

Please also suggest a visual abstract to illustrate your article as a PNG file 550 px wide x 300-600 px high. A cropped portion of this image will serve as thumbnail for the table of content on our webpage.

16) As part of the EMBO Publications transparent editorial process initiative (see our Editorial at <http://embomolmed.embopress.org/content/2/9/329>), EMBO Molecular Medicine will publish online a Review Process File (RPF) to accompany accepted manuscripts.

This file will be published in conjunction with your paper and will include the anonymous referee reports, your point-by-point response and all pertinent correspondence relating to the manuscript. Let us know whether you agree with the publication of the RPF and as here, if you want to remove or not any figures from it prior to publication.

I look forward to receiving your revised manuscript.

With kind regards,

Lise

**** Reviewer's comments ****

Referee #1 (Remarks for Author):

In the revised manuscript, the authors have clarified a number of findings that were described as correlations in the first draft. I had asked the authors to study causality between the observations, which has now been done to my satisfaction. Especially the finding that SMOC2 is required for the phenotype is compelling.

Referee #2 (Comments on Novelty/Model System for Author):

The authors implemented the manuscript with advanced techniques producing significant results. This allowed to obtain new and relevant data.

Referee #2 (Remarks for Author):

I particularly appreciated the work of the authors who have significantly improved the quality of the manuscript in this version. The new obtained results complete the description of the biochemical and molecular mechanism of the interaction between KRAS and EGFR mutations. The authors have responded to all criticisms in a clear and complete manner, also performing experiments of considerable complexity and commitment. My opinion is that the manuscript in this version is certainly worthy of publication.

Referee #3 (Comments on Novelty/Model System for Author):

I do not think the the data as presented will represent any relevant advance in the field of cancer therapy.

Referee #3 (Remarks for Author):

In this revised version of the manuscript, the authors have included additional information that strengthens some aspects of the work. However, I still feel that there is a lack of connection between the different sections and the results obtained from them. Specifically, while it is true that the metabolism section is now more complete, it is surprising to me that glut1 emerges as a relevant element that is downregulated upon EGFR depletion and is responsible for a reduction in glucose uptake, but is then ignored. Additional experiments showing the impact of GLUT1 downregulation in the AKPE model would be informative. Furthermore, it is mentioned that only membrane-bound GLUT1 is reduced in AKPE cells, but not the nuclear fraction. This needs to be demonstrated by WB or IF analysis. Furthermore, it has been suggested previously that different isoforms are responsible for the different subcellular distribution of GLUT1. Did the authors determine whether AKP and AKPE cells express different isoforms of the glut1 gene?

After performing RNA-seq, the authors always mention the AKPE signature, but what this signature includes is never well defined. Is the AKPE signature all the genes that are differentially expressed in AKPE vs. AKP? In this case, since there are up-regulated and down-regulated genes, how is the high or low signature defined? Understanding what the AKPE signature means high and low is relevant to several of the conclusions drawn, including those derived from EV6A or Figures 8D and 8E.

Related to Figure 5, where the authors focus on the observation that AKPE organoids are smaller than AKP organoids, although I also find it interesting to look at cell size and the idea that AKPE cells are smaller because they are more stem-like, a reduction in organoid diameter would immediately suggest a reduction in proliferation, especially if the pathways downregulated upon EGFR deletion include MAPK and cell cycle (Figure 3E).

The authors should look at ki67 and cell cycle in these organoid models.

In Figure 5E, SMOC2 depletion in the organoids does not reduce the transcriptomic distance between AKPE and AKP, as I would have expected. This unexpected (?) result needs to be mentioned and discussed. Not only that, but if SMOC2 levels are almost undetectable in AKP organoids, why does SMOC2 KO in these cells result in a high separation in PCA?

Since the authors focus in Figure 1 on GLUT1, which is post-transcriptionally regulated upon EGFR deletion and is a relevant mediator of changes in glucose uptake, it should be investigated whether GLUT1 and membrane-bound GLUT1 are rescued upon SMOC2 deletion in the AKP.

In the single cell RNA-seq of AKP and AKPE, I don't really see the higher heterogeneity of AKP cells that the authors mention. What I see is a continuum of cell populations in the AKPE organoids with high transcriptional dispersion, even higher than in the AKP. Perhaps the sub-clustering would be more convincing if it were done in the UMAP, which includes both genotypes together.

Figure 7 shows that EGFR deletion and MAPK inhibition both contribute to reduced proliferation and there is a clear additive effect when both elements are removed. However, I do not see any evidence or formal demonstration of synergy as stated in the subheading of the section.

In Figure 8 and in relation to the Kaplan-Meier analysis based on the AKPE signature, I already mentioned that it is essential to define what high and low means. Also, I'm very surprised to see that the signature imposed by EGFR depletion is of good outcome only in the KRAS mutated tumors. Does this mean that EGFR inhibition has no effect in tumors with WT KRAS? in my opinion, these results need to be specifically mentioned and discussed because they have a clear clinical relevance.

In conclusion, I think that there are a lot of relevant data in the manuscript, but the different observations are sometimes confusing and the results are not always followed up in the different sections of the manuscript.

Remaining concerns:

In particular, we would like you to address experimentally the following points:

- Demonstration that only membrane-bound GLUT1 is reduced in AKPE cells, but not the nuclear fraction OR tone down this claim.

These data are already in the manuscript: membrane-bound GLUT1 quantification was assessed by IF and depicted in Fig 1F (see also below). Membrane GLUT1 is reduced and nuclear GLUT1 unchanged in AKPE organoids. We have toned down this claim in the results section (page 7, line 147, 149) as p value is 0.0506. Also discussion was adapted (page 21, line 555).

- Figure 5, AKPE organoids smaller than AKP organoids: Ki67 staining and/or cell cycle analysis.

Also these results (again criticized by Referee #3) are already shown in the manuscript. We were similarly puzzled and analysed proliferation by different methods, namely Ki67 and EdU shown in Fig. 1C (see below). Additionally, also the expression of cell cycle genes were shown by RNASeq in EV3A and reconfirmed by an independent RNASeq analysis, depicted in Fig. 7D, AKPE vs. AKP).

- single cell RNA-seq of AKP and AKPE: sub-clustering in UMAP.

We have added an additional UMAP representation colored by the AKP and AKPE subclusters (Fig. 6B). The original UMAP subclustering, previously shown in 6B, is now moved to EV6A.

- Figure 7: formal demonstration of synergy OR tone-down.

We have toned down this claim on page 17, line 435; p18, line 493-494.

13th Mar 2025

Dear Maria,

Thank you for submitting your revised files. I am glad to say that I will be able to accept your manuscript once the following minor editorial issues are addressed:

1/ Manuscript text:

- Please remove the underlined text, and only keep in track changes mode any new modification.
- Discrepancies in authors' names were found: Emi Adachi-Fernandez in the manuscript file vs. Isabel-Emi Adachi Fernandez in the submission system; Cristiano de Sá Fernandes in the manuscript file vs. Alfredo Cristiano De Sa Fernandes in the submission system. Please correct as needed.
- We can accommodate a maximum of 5 keywords, please adjust accordingly.
- "Materials and Methods" should be renamed "Methods":
 - o Please provide a Reagents and Tools table (listing key reagents, experimental models, software and relevant equipment and including their sources and relevant identifiers). Please download and fill our Reagents and Tools Table template (.docx), which you can find in our author guidelines: <https://www.embopress.org/page/journal/14693178/authorguide#structuredmethods>. When submitting your revised manuscript, please do not include the Reagents and Tools Table in the Methods section of the manuscript but upload it as a separate file choosing the file type "Reagent Table".
 - o Animals: please provide information on mouse strains, as well as mice gender and age at time of experiments.
 - o Antibodies: please provide dilutions/concentrations.
 - o Please provide a statement on mycoplasma contamination.
 - o Statistics: please provide statements on sample size, exclusion/inclusion criteria, blinding and randomization.
- Please remove the section on "Material availability".
- Funding should be part of the Acknowledgements.
- References: Please note that the data citations are not tagged with the label "DATASET" in the reference list.

2/ Figures and Appendix:

- Please note that you have the possibility to make your Appendix figures EV figures, as we can now accommodate more than 5 EV figures. If you choose to keep an Appendix, please note that the names of the figures should be updated to Appendix S1 and Appendix S2.
- Datasets EV1, EV2 and EV5 appear like regular tables, and should be made main or EV tables.
- Please make sure to describe the statistical test used and exact p values throughout the figures/figure legends.
- Please define the scale for Figure 2E (as it is too small to read in the picture).

3/ Source data:

Thank you for providing Source Data. Please complete with uncropped blots and images as requested in the checklist, and group them with the numerical files as one (zipped) file per figure.

4/ Checklist:

- please fill in the subsection on mycoplasma contamination.
- please fill in all subsection of "Experimental study design and statistics".
- please check whether you need to fill the subsection on "specimen and field samples", as I don't think this applies to your study.

5/ Thank you for providing The paper explained, please include it in the main manuscript text file.

6/ Synopsis:

Thank you for providing a nice visual abstract. Kindly upload it as an individual jpeg/TIFF/png file, 550 pixels wide x 300-600 pixels high. A small portion will be cropped to serve as thumbnail on our webpage.

I introduced minor edits in your synopsis text to fit with our style and format, please let me know if you agree with the following or amend as you see fit:

"The role of EGFR role in KRASG12D-driven tumorigenesis was studied using mouse CRC organoids with KRAS-APC-TP53 mutations and inducible EGFR deletion.

- EGFR deletion in KRASG12D mutant organoids shifted metabolism from glycolysis to glutaminolysis.
- Loss of EGFR downregulated MAPK, PI3K, and ErbB pathways and induced a cancer stem cell WNT/stemness signature with reduced cell size.
- Smoc2 was identified as a key target mechanistically driving these phenotypes.

- Reduced phenotypic heterogeneity was observed in organoids lacking EGFR.
- The identified transcriptional signature correlated with improved survival in KRAS mutant CRC patients."

7/ As part of the EMBO Publications transparent editorial process initiative (see our Editorial at <http://embomolmed.embopress.org/content/2/9/329>), EMBO Molecular Medicine will publish online a Review Process File (RPF) to accompany accepted manuscripts.

This file will be published in conjunction with your paper and will include the anonymous referee reports, your point-by-point response and all pertinent correspondence relating to the manuscript. Let us know whether you agree with the publication of the RPF and as here, if you want to remove or not any figures from it prior to publication.

I look forward to receiving your revised manuscript.

With kind regards,

Lise

The authors addressed the remaining editorial issues.

2nd Apr 2025

Dear Maria,

Thank you for submitting your revised files. I am pleased to inform you that your manuscript is accepted for publication and is now being sent to our publisher to be included in the next available issue of EMBO Molecular Medicine!

With kind regards,

Lise
